# TiC-LM: A Multi-Year Benchmark for Continual Pretraining of Language Models

## Abstract

Large language models (LLMs) are trained on data crawled over many years from the web. We investigate how quickly LLMs become outdated as the world evolves with time and how to best update them with newer data. Specifically, we simulate a world where the latest dump of Common Crawl (CC), the most prominent public source of pre-training data, is used every month to *continually* train an LLM. We design various dynamic evaluations from the CC data, Wikipedia, StackExchange, and code documentations to measure continual learning metrics such as forgetting and forward transfer. Notably, our TiC-CC training data is more than $100\times$ larger compared with prior continual learning benchmarks for language modeling. We discover that recent DataComp-LM (Li et al., 2024a) models trained on data before 2023 have already become outdated, incurring up to 45% larger noun-perplexity on 2024 Wikipedia articles compared to pre-2023 articles. Further, we use our setup to evaluate the effectiveness of several large-scale continual learning methods and find that replaying older data is most effective for combating forgetting: for previously seen CC dumps, it can reduce the regret on held-out loss by 60% compared to other optimizer and loss-based interventions. However, some domains evolve more quickly than others, favoring different trade-offs between mixing old and new data.

## 1 Introduction

Large language models (LLMs) rely on massive amounts of data, a major portion of which comes from large-scale web-crawls that have been running over the past 10–20 years. Common Crawl (CC), the most well-known source of such data, has been active since 2007 and continues to release monthly dumps of data. While typically many (or all) previous dumps are combined together to train LLMs from scratch (Penedo et al., 2023; Li et al., 2024a), the vast costs and inherent knowledge cutoffs of LLMs raise natural questions about how they can be most effectively updated as future dumps are released. In this work, we introduce a benchmark for Time-Continual Learning of Language Models (TiC-LM) and investigate how to continually train LLMs over many months and years. Taking inspiration from the recent TiC-CLIP (Garg et al., 2024) work, our goal is to find efficient alternatives to training LLMs from scratch by reusing and updating prior pre-trained models. Overall, we seek to answer the following:

- How quickly does a pre-trained LLM become outdated? We observe that Gemini, Gemini-2, and DCLM models are outdated on 2024 data by 34%, 28%, and 45%, respectively (Fig. 2).
- Can continual pretraining reach the performance of training from scratch when fixing the number of tokens? We demonstrate a variety of methods that shrink the gap, but this proves to be an open and challenging problem to be studied in future using our benchmark.
- Do forgetting and forward transfer vary across domains such as Wikipedia, News, etc? Yes. For example, although data replay methods generally avoid forgetting, they hurt performance on domains that change rapidly (Fig. 5).

Our contributions in TiC-LM are: (1) introducing a large-scale continual pretraining benchmark for language modeling, and (2) evaluating continual learning strategies. TiC-LM centers around TiC-CommonCrawl (TiC-CC), a massive time-stratified set of training and evaluation data created using 114 CC dumps spanning 2013–2024 including evaluation subsets TiC-CC-Wiki and TiC-CC-News. TiC-CC contains 2.9T tokens, more than $100\times$ larger than prior continual continual learning benchmark for LLMs. TiC-LM also contains domain-specific evaluations sourced from

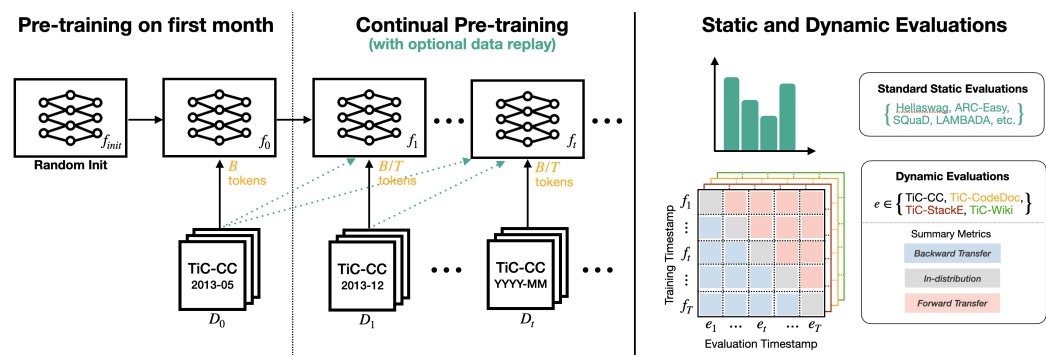

Figure 1: **TiC-LM experiment setup.** We simulate a setup where each Common Crawl dump $D_0, \cdots, D_T$ is revealed one-at-a-time. An LLM $f_0$ is first pre-trained for $B$ tokens on the initial month $D_0$ and then continually updated for a fixed budget of $B/T$ tokens in each following month (which may optionally include replaying any older data). The goal is for each monthly model $f_1, \cdots, f_T$ to perform well on both standard static downstream tasks as well as dynamic evaluations that evolve over time, requiring the balancing of learning (gray/red) with preventing forgetting (blue).

Table 1: Comparison with continual learning benchmarks for LLMs. CLS: classification, SUM: Summarization KB: Knowledge Base, QA: Question-Answering, LM: Language Modeling. Acc.: Accuracy, Ppl.: Perplexity, Tok.: Tokens, Art.: Articles.

| Benchmark | Domain | Task | Metric | CL Train | Time-CL | Years | Timesteps | # CL Train | # Eval Samples |
|---|---|---|---|---|---|---|---|---|---|
| Gururangan et al. (2020) | Science,News,Reviews | CLS | micro/macro-F1 | ✓ | ✗-Task CL | — | — | 0.3M | 140k |
| Luu et al. (2022) | Tweet,Science,News,Reviews | CLS/SUM | F1/Rouge-L | ✓ | ✓ | 2013–2022 | 4–7 | 695k | 695k |
| Chrono. Tweet(2022) | Science,Tweet | CLS | micro/macro-F1 | ✓ | ✓ | 2014–2020 | 4 | 25M | 4M |
| TempEL (2022) | Wikipedia | KB | EL Acc. | ✓ | ✓ | 2013–2022 | 10 | — | 92k |
| TemporalWiki (2022a) | Wikipedia | KB | Noun Ppl. | ✗ | ✓ | 2021 | 4 | 23B Tok. | 36k |
| StreamingQA (2022) | News | QA | Acc. | ✓ | ✓ | 2007–2020 | 12 | 99k Art. | 46k |
| EvolvingQA (2024) | Wikipedia | QA | EM/F1 | ✓ | ✓ | 2007–2020 | 6 | — | 46k |
| TIQ (2024) | Wikipedia | QA | Precision/Rank | ✓ | ✓ | 1801–2025 | — | 6k QA | 4k |
| TAQA (2024) | Wikipedia | QA | F1 | ✓ | ✓ | 2000–2023 | — | 9k QA | 11k |
| TIC-CC (All/Wiki/News) | Generic Web | LM | Ppl. | ✓ | ✓ | 2013–2024 | 114 | 2.9T Tok. | 2.7M |
| TIC-WIKI | Wikipedia | KB | Noun Ppl. | ✗ | ✓ | 2014–2024 | 62 | — | 10M |
| TIC-STACKE | Code,Math,English,... | KB / QA | Ppl. | ✗ | ✓ | 2008–2024 | 187 | — | 3.65M |
| TIC-CODEDOCS | Code | LM | Ppl. | ✗ | ✓ | 2017–2024 | 16 | — | 6.5k |

outside Common Crawl including TiC-Wikipedia (TIC-WIKI), TiC-StackExchange (TIC-STACKE), and TIC-CODEDOCS spanning 2008–2024. Using our benchmark, we evaluate several continual learning baselines and find that cyclic learning rate schedules and data replay can be effective for balancing learning on new data and preventing forgetting. We also find that different domains evolve at different rates, favoring different methods (e.g., benefiting from more or less replay).

## 2 RELATED WORK

Learning new capabilities from multiple, sequentially observed, distributions has long been an active area of ML research (Wang et al., 2024). More recently, several works have studied this setting for LLMs (Wu et al., 2024), targeting improvements on: (1) general capabilities (via updating on improved datasets) (Parmar et al., 2024; Ibrahim et al., 2024; Gupta et al., 2023); (2) specific domains (Jin et al., 2022; Gururangan et al., 2020; Chen et al., 2024); (3) newer data as the world evolves (Jin et al., 2022; Jang et al., 2022b;a; Lazaridou et al., 2021; Nylund et al., 2024; Loureiro et al., 2022; Qin et al., 2022; Liška et al., 2022). Works in this third category have demonstrated that in many domains, the performance of LLMs decay as training and evaluation sets grow farther apart in time, motivating the need for methods to *efficiently* and *non-distruptively* adapt to temporal distribution shifts. Our work scales up these previous efforts to more closely match current LLM training practices. While older works typically focus on continual training runs involving individual sources (e.g., news, Wikipedia, and social media) and <10 timesteps, we consider training on a *generic web-crawl* (i.e., Common Crawl) spanning 114 different months. In turn, the generality of our training data allows us to go beyond single-domain evaluations. We provide an extended discussion of related works in Appx. E. Table 1 summarizes our proposed datasets compared with the most related time-continual benchmarks. With 2.9T tokens, TIC-CC is the *largest and most diverse* continual learning benchmark for language modeling.

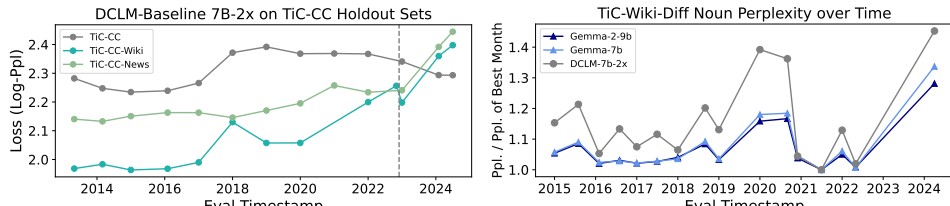

Figure 2: (Left) Performance of a model trained on DCLM-Baseline, which contained data up to 2022. Notably, the loss increases significantly on TIC-CC-WIKI and TIC-CC-NEWS subsets after 2022 data cutoff. (Right) Performance of the same DCLM-Baseline model as well as two versions of Gemma (GemmaTeam et al., 2024) on our TIC-WIKI dynamic evaluation. Performance of the DCLM model is about 45% worse on the latest evaluation data compared to the preceeding data. For the Gemma series, the older Gemma-7b and newer Gemma-2-9b are 34% and 28% worse, respectively.

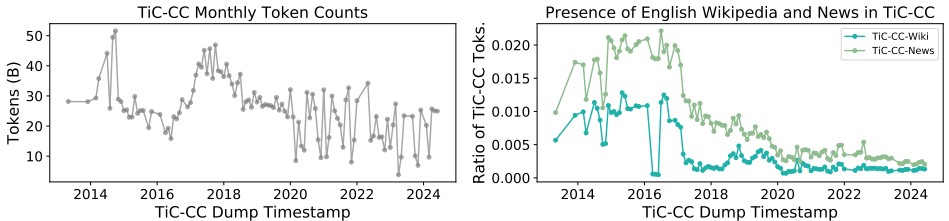

Figure 3: We plot the total number of tokens per month in TIC-CC (left) as well as the proportion of those tokens coming from our TIC-CC-WIKI and TIC-CC-NEWS subsets (right).

## 3 TIC-COMMONCRAWL (TIC-CC): MORE THAN A DECADE OF WEB DATA

We create a large *time-stratified* dataset of 2.9T tokens based upon Common Crawl, a free and open corpus of web-crawled data that has been online since 2007. CC releases new snapshots of the web roughly every month. Each dump creates a representative snapshot of the web by sampling a limited number of pages from each domain. Sampled URLs change from month to month independent of whether they appeared in the previous dumps. We collect all dumps between May-2013 and July-2024, resulting in 114 corresponding splits that we refer to by the month of their release date. For each split, we then apply a pre-processing pipeline based on that of DataComp-LM (Li et al., 2024a). Notably, we do not perform any operations on a particular month that depend on future months to retain causality and temporal order.

**Data processing.** We build upon the existing pipeline from DataComp-LM (Li et al., 2024a), starting with DCLM-Pool (Li et al., 2024a), which contains all CC dumps between May-2013 and Dec-2022 and parsed to extract plain text from webpages via the open-source `resiliparse` library (Bevendorff et al., 2018; 2021). We split this data by month and reuse the same download and processing scripts to extend DCLM-Pool until July-2024. Next, we follow DCLM-Baseline's pipeline by applying heuristic filters from RefinedWeb (Penedo et al., 2023) and a fuzzy-deduplication step which we modify to run only *within* each month rather than non-causal global deduplication. Alternatively, similar to TiC-CLIP, one could deduplicate data globally but keep the earliest occurrence of each document. We avoid this deduplication for two reasons: (1) fuzzy deduplication across months may not always be helpful, potentially removing near-duplicates such as Wikipedia pages where a few key facts have changed but most of the text is the same, (2) it allows for exploring the benefits/pitfalls of such data-centric interventions as part of method design. Also, we do not use the final classifier-based filter in DCLM-Baseline, as this classifier was trained on data from all months.

Finally, we leverage the fact that DCLM-Pool was randomly partitioned into ten equally-sized chunks to construct held-out sets for loss-based evaluations (Sec. 4.1). In Fig. 3, we show the number of tokens we have for each month of the dataset. In total, the dataset spans 29T tokens, with individual months ranging between 100B to 500B tokens. We use smaller subset of 220B tokens from a single global shard with 2.9T for our training while future work can expand to the full 2.9T/29T tokens. For more details about the data pipeline see Appx. A.

## 4 EVALUATIONS

In this section, we will discuss various time-continual evaluations that are designed both with and independent of CC data. As our focus is on continual pretraining, we focus on evaluations without instruction-tuning. We introduce three sets of novel evaluations: TIC-CC, TIC-CC-WIKI, TIC-CC-NEWS, TIC-WIKI, TIC-STACKE, and TIC-CODEDOCS.

**Static downstream evaluations.** We focus on pre-trained base models without any instruction fine-tuning and evaluate our models on a variety of suitable downstream zero-shot and few-shot tasks. Specifically, we use the CORE evaluations from the DCLM benchmark (Li et al., 2024a) which includes 22 zero-shot and few-shot in-context learning tasks. These evaluations, which include benchmarks such as ARC-Easy (Clark et al., 2018) and Hellaswag (Zellers et al., 2019), assess general capabilities of base models via a variety of world knowledge and natural language understanding tasks. While these evaluations are not designed to be time-dependent, we use them to assess (1) whether continually trained models match the general capabilities of models trained on all dumps; (2) if they benefit from different months of Common Crawl.

**Perplexity metrics.** We employ three distinct perplexity metrics for different evaluations:

$$\text{ppl}_{\text{token}} = \exp\left(\frac{\sum_{d \in \mathcal{D}} \sum_{t \in T_d} -\log P(t|c_{<t})}{\sum_{d \in \mathcal{D}} |T_d|}\right), \tag{1}$$

where $\mathcal{D}$ is a set of documents, $T_d$ is the set of tokens in document $d$, and $c_{<t}$ is the context prior to token $t$.

$$\text{ppl}_{\text{answer}} = \frac{1}{|\mathcal{Q}|} \sum_{q \in \mathcal{Q}} \exp\left(-\log P(a_q|c_q)\right), \tag{2}$$

where $\mathcal{Q}$ is a set of question-answer pairs, $a_q$ is the gold answer for question $q$, and $c_q$ is the context.

$$\text{ppl}_{\text{noun}} = \exp\left(\frac{\sum_{d \in \mathcal{D}} \sum_{n \in N_d} -\log P(n|c_{<n})}{\sum_{d \in \mathcal{D}} |N_d|}\right), \tag{3}$$

where $\mathcal{D}$ is the set of documents in a snapshot, $N_d$ is the set of proper noun tokens (tagged as NNP or NNPS by a POS tagger) in document $d$, and $c_{<n}$ is the context prior to noun $n$.

### 4.1 TIC-COMMONCRAWL (TIC-CC) EVALUATIONS

CC data is a consistent, albeit partial, snapshot of the web over years that does not require special processing of the history per website. We compute token-perplexity ($\text{ppl}_{\text{token}}$) on three monthly subsets of our CC data which were held out from training:

- TIC-CC: Held-out documents coming from the full distribution for each month of TIC-CC.
- TIC-CC-WIKI: Pages in TIC-CC from English Wikipedia (i.e., whose URLs contain either the domain `en.wikipedia.org` or `simple.wikipedia.org`).
- TIC-CC-NEWS: Pages in TIC-CC from a set of news sites based on WMT competitions (Barrault et al., 2020).

### 4.2 TIC-WIKIPEDIA (TIC-WIKI)

Our TIC-CC-WIKI evaluation in Sec. 4.1 is based on sampled Wikipedia pages existing in each CC dump which is a representative set of Wikipedia but not all of it. We create TIC-WIKI, a more comprehensive evaluation from full dumps of Wikipedia while utilizing the knowledge graph from Wikidata. TIC-WIKI allows us to construct question/answer and factual evaluations as well as split the performance over changed/unchanged knowledge. We build upon TemporalWiki (Jang et al., 2022a), which generates evaluations from four consecutive monthly snapshots of English Wikipedia and Wikidata. Our TIC-WIKI evaluation captures a broader spectrum of knowledge evolution, we extend the evaluation timespan to a full decade (2014–2024) and improve upon the matching of Wikipedia/Wikidata (see Appx. B.1 for more details).

To evaluate performance on TIC-WIKI diffsets, we adopt the approach of Lazaridou et al. (2021) and Jang et al. (2022a), calculating the average perplexity of proper nouns ($\text{ppl}_{\text{noun}}$) identified by a Part-of-Speech tagger (Honnibal & Montani, 2017). This method serves as a proxy for assessing factual knowledge changes, as proper nouns often contain key factual information.

## 4.3 TIC-STACKEXCHANGE (TIC-STACKE)

We design another question/answering evaluation based on the historical data from StackExchange. StackExchange has 182 communities that share knowledge by posting questions and answers. We measure answer-perplexity ($ppl_{answer}$) on high-quality answers from selected sites by collecting answers that have been accepted by the question author (using the accepted answer timestamp to bin examples by month). The resulting evaluation contains examples from 2008–2024. We provide details of TIC-STACKE data processing in Appx. B.2.

## 4.4 TIC-CODE DOCUMENTATIONS (TIC-CODEDOCS)

Our TIC-CODEDOCS evaluation is based on code documentations from popular open-source Python libraries: NumPy (Harris et al., 2020) and PyTorch (Ansel et al., 2024). For NumPy, we use documentations from 16 major releases, ranging from version 1.13.0 (June 2017) to version 2.1.0 (August 2024). For PyTorch, we use documentations of major releases ranging from version 1.8.0 (March 2021) to version 2.4.0 (July 2024).

We build the documentation directly from each library's git repository. The process involves the following steps: (1) Identify the commit tagged for the major release, (2) revert to that specific commit, (3) install necessary dependencies, (4) build the project from source, (5) generate HTML documentation from the source. We then convert all HTML pages to raw text by extracting the main body of the documentation pages. This approach ensures that template-related elements of the pages such as the index and footer are not included in the final text, focusing solely on the relevant documentation content. We evaluate the model's code understanding using perplexity ($ppl_{token}$), calculated across entire snapshots of code docs.

## 5 CONTINUAL LEARNING BASELINE METHODS

The goal for TiC-LM methods is to match the performance of the *Oracle* which trains on all data (114 months) starting from random initialization for the full token budget. We consider methods from three categories: optimization-based, data replay, and regularization. Aside from average metrics across all timesteps, methods should balance forgetting and forward transfer metrics (defined in Sec. 6).

**Optimization-based methods.** In non-continual settings, LLMs are often trained with a cosine-decayed learning rate schedule which requires knowledge of total training steps ahead of time. In a continual setup, however, the number of total tokens grows over time and we care about the performance after each month. We benchmark the following optimization approaches in our work:

- *Cyclic Cosine* is the simplest alternative which applies cosine decay *within* each training month using the same maximum learning rate and warmup for each round. This was found to be most effective in TiC-CLIP (Garg et al., 2024).
- *Cyclic Cosine + AR (autoregressive)* is similar to cyclic cosine decay except the maximum learning rate in each cycle decays across months, regressed from a single-cycle cosine decay and shown to offer improvements by Roth et al. (2024).
- *Rsqrt (reciprocal-$\sqrt{}$)* are *infinite* schedules that decay the learning rate slowly in a global training run and branch off of this trajectory with linear cooldowns (Zhai et al., 2022). To keep training steps fixed compared to other methods, we follow Roth et al. (2024) and implement a version that maintains only a single trajectory by re-warming up from the previous cooldown.
- *Schedule-Free* is an optimizer proposed by Defazio et al. (2024) which aims to circumvent the need for defining a learning rate schedule by using iterate averaging and has achieved promising results in i.i.d. non-continual settings.

**Data replay methods.** Data replay is a classical continual learning strategy to prevent forgetting, whereby in each training round, the model is fed a mixture of data from both older and the current timesteps (Lopez-Paz & Ranzato, 2017; Rebuffi et al., 2017; Chaudhry et al., 2018). Defining a replay method therefore boils down to *how* the mixture ratios are specified. We consider the following replay strategies based on the best-performing strategies in TiC-CLIP (Garg et al., 2024):

- For the current timestep $t$, we allocate a ratio $0 \leq \alpha_t \leq 1$ of the monthly token budget $B_t$ to data from the current month, seeing $\alpha_t B_t$ tokens from that month.

- For previous months, we redistribute the remaining $(1 - \alpha_t)B_t$ tokens equally, i.e., each month contributing $\frac{1-\alpha_t}{t-1}B_t$ tokens to this round's training set.

In particular, when $\alpha_t = 1/t$, we see an equal number of tokens from all observed months. We also consider setting $\alpha_t = 1/2$ which always allocates half the token budget to the current month. The general downside of replay-based methods is the cost of retaining old data. This can be particularly challenging if old data expires and needs to be removed. Methods with larger values of $\alpha_t$ are less affected by such limitations.

**Regularization-based methods.** These methods alter the training objective instead of the data, generally by adding a regularization term which encourages newer model updates to stay close to the model weights learned after the previous month. Following TiC-CLIP, we try two notable methods: LwF (Li & Hoiem, 2018) and EWC (Kirkpatrick et al., 2017).

- *LwF* adds an additional loss term based on KL divergence which penalizes differences in model outputs between the previous checkpoint and the current model.
- *EWC* attempts to slow down updates to particular model parameters which are highly influential for performing well on older months as measured by the (approximate) Fisher information matrix.

Because both LwF and EWC involve extra loss terms and model copies, it is important to note that they induce larger GPU memory footprints and run-times compared to optimizer and replay-based methods. That being said, we do not try to adjust the token counts to account for this given that our re-implementations may not be optimally efficient.

## 6 EXPERIMENTS

**Training details.** For all runs, we train 3B parameter language models using OpenLM. Unless otherwise indicated, each method observes a fixed total number of 220B tokens, equivalent to 4x the Chinchilla optimal amount. [1] We further assume that current practitioners are (a) likely to have access to more than enough data to train initial models; (b) unlikely to wait to obtain non-trivial performance. Hence, we front-load the total token budgets such that *half* is allocated to training on the first month, May-2013. Then, the remaining 110B tokens are split equally among the other 113 continual timesteps. For more realistic hyperparameter selection in our continual setup (Cha & Cho, 2024), we only use the first 10 of these timesteps for tuning (see Appx. C for more details).

**Evaluation metrics.** Each continual run actually produces a $T_t \times T_e$ matrix of evaluations $E$ where $T_t, T_e$ are the total number of training/evaluation timesteps, $E_{i,j}$ is the performance of the model trained after training on data up to month $i$ and evaluated on the month $j$. To control for inherent difficulty differences across evaluation months, we measure the *regret* $R_{i,j} = E_{i,j} - E_j^*$ where $E_j^*$ is the performance of the *Oracle* trained on all months (May-2013–July-2024) on month $j$. We subtract $E_j^*$ instead of $E_{j,j}$ since if $E_{j,j}$ is bad it may lead to misleadingly good forward/backward metrics.

Following Garg et al. (2024), we consider the following summary metrics (assuming $T_t = T_e = T$).

- In-distribution (ID) performance: averages along the matrix diagonal, i.e., $\sum_{i=1}^{T} = R_{i,i}/T$.
- Backward transfer: averages the lower triangular of $R$, i.e., $\sum_{i=1}^{T} \sum_{j<i} \frac{R_{i,j}}{T(T-1)/2}$.
- Forward transfer: averages the upper triangular of $R$ analogously to backward transfer.

For some downstream evaluations, the train/evaluation periods do not exactly align ($T_t \neq T_e$), making the definition of ID more nuanced. For such evaluations, we define $a_i$ as the index of the nearest evaluation timestep that comes before the training timestep $i$. We then count $R_{i,a_i}$ towards the ID average only if no other training timestep is closer to $a_i$ (i.e., $a_i \neq a_{i-1}$). Otherwise, we count $R_{i,j}$ towards backward and forward transfer when $j < a_i$ and $j \geq a_i$ respectively.

### 6.1 TIC-CC HELD-OUT EVALUATIONS

Tab. 2 and Fig. 4 show results on our TiC-CC hold-out sets. Overall, we observe the various methods incur different trade-offs between forgetting and plasticity. We summarize the main findings below:

---

[1] Here, following Li et al. (2024a), the token counts are given by $20 \times$ number of parameters $\times$ Chinchilla multiplier with a $1\times$ multiplier being a near-optimal compute allocation found by Hoffmann et al. (2022).

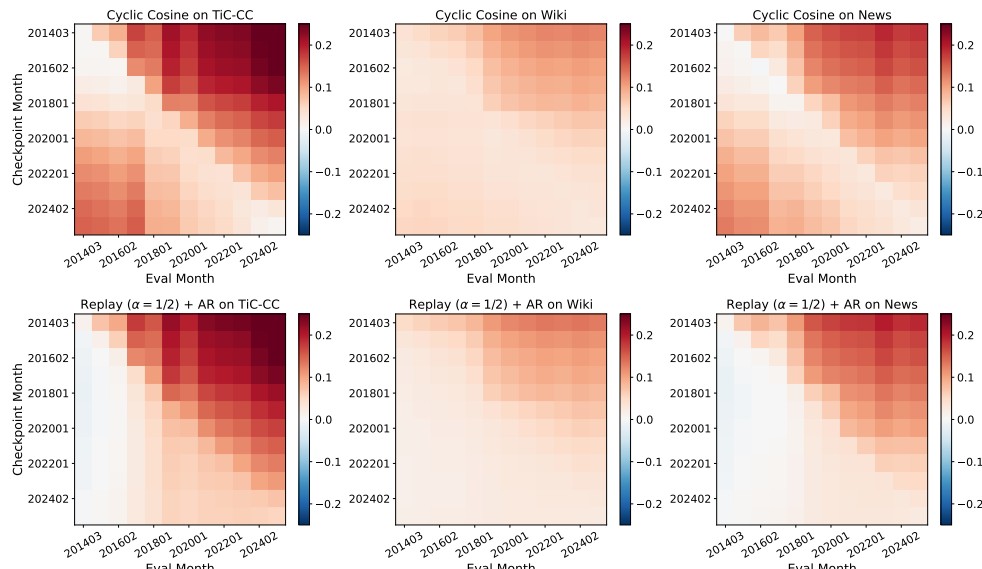

Figure 4: **Cyclic Cosine demonstrates delayed forgetting while replay avoids forgetting in exchange for lower plasticity.** We plot the difference in log-perplexity $(\log \text{ppl}_{\text{token}})$ between continual checkpoints and the Oracle. While we train on all 114 months, we evaluate on a subset of the months which are roughly annually spaced. Overall, we observe that training sequentially with the Cyclic Cosine method (top) leads to strong ID performance (along the diagonal) but also significant forgetting on TIC-CC and TIC-CC-NEWS. Meanwhile, adding replay can reduce this forgetting by sacrificing ID performance, especially for later model checkpoints.

Table 2: **Loss-based evaluations for various methods at the 3B-4x scale.** We report log-perplexity values relative to the Oracle. While various optimizer (top) and regularization-based (bottom) methods trade-off backward transfer with in-distribution performance, replay (middle) is required to obtain the least amount of forgetting. **Bold** values are within one standard deviation of the best in each column, with standard deviations estimated from three runs of Cyclic Cosine.

| Method | TIC-CC ↓ | | | TIC-CC-WIKI ↓ | | | TIC-CC-NEWS ↓ | | |
|---|---|---|---|---|---|---|---|---|---|
| | Backward | ID | Forward | Backward | ID | Forward | Backward | ID | Forward |
| Cyclic Cosine (std) | 0.072 (0.000) | **0.027** (0.000) | **0.161** (0.000) | 0.038 (0.000) | 0.032 (0.000) | 0.074 (0.000) | 0.058 (0.000) | 0.015 (0.000) | 0.109 (0.000) |
| Cyclic Cosine + AR | 0.058 | 0.040 | 0.166 | 0.032 | 0.031 | 0.074 | 0.041 | 0.017 | 0.110 |
| Cyclic Rsqrt | 0.065 | 0.030 | 0.162 | 0.033 | 0.030 | 0.073 | 0.049 | 0.015 | **0.108** |
| Schedule-Free | 0.065 | 0.036 | 0.164 | 0.036 | 0.033 | 0.076 | 0.049 | 0.017 | 0.110 |
| Replay ($\alpha_t = 1/t$) | **0.023** | 0.074 | 0.178 | 0.020 | 0.036 | 0.078 | 0.005 | 0.035 | 0.117 |
| Replay ($\alpha_t = 1/2$) | 0.024 | 0.042 | 0.167 | 0.024 | 0.031 | 0.074 | 0.013 | 0.019 | 0.111 |
| Replay ($\alpha_t = 1/t$) + AR | 0.026 | 0.083 | 0.181 | **0.019** | 0.037 | 0.079 | **0.004** | 0.039 | 0.119 |
| Replay ($\alpha_t = 1/2$) + AR | 0.025 | 0.055 | 0.171 | 0.022 | 0.032 | 0.076 | 0.009 | 0.022 | 0.112 |
| LwF | 0.072 | **0.027** | **0.161** | 0.038 | 0.032 | 0.074 | 0.058 | 0.015 | 0.109 |
| EWC | 0.061 | 0.032 | 0.162 | 0.031 | **0.029** | **0.071** | 0.046 | **0.014** | **0.108** |

**Cyclic Cosine offers the best plasticity but also the most forgetting.** We find the optimal maximum learning rate for ID performance to be $30\times$ smaller than what was used for the initialization (see Tab. 5 in Appx. C). Further, based upon the gaps between Forward and ID metrics, TIC-CC-WIKI *appears to change more slowly than* TIC-CC-NEWS*, while both evolve less rapidly than* TIC-CC.

**Alternative LR schedules and EWC can improve Backward at the cost of ID but replay is *required* to further reduce forgetting.** As shown in the heatmaps in Fig. 12 , all non-replay methods still result in significant forgetting at later checkpoints, while Replay ($\alpha_t = 1/t$) reaches a Backward metric of 0.023 on TIC-CC, 60% smaller than the best non-replay approach. Between the two replays, $\alpha_t = 1/2$ offers slightly less Backward improvement but much better ID, likely because $\alpha_t = 1/t$ decreases the ratio of new data over time, scaling poorly to the >100 timesteps in our setup. This differs from TiC-CLIP's findings, where different replays behave more similarly but $10\times$ fewer rounds are used. However, even while $\alpha_t = 1/2$ can achieve good balance between Backward and ID for the first few years, plasticity becomes an increasing issue across the larger timescales in TIC-CC, likely due to later months still being underrepresented as they have been replayed fewer times.

Table 3: **Selected downstream evaluations at 3B-4x scale.** For all dynamic evaluations, we report perplexity values relative to the Oracle with log-scaling. Meanwhile, CORE is an average of the accuracies of 22 downstream zero/few-shot tasks used in DataComp-LM (Li et al., 2024a), evaluated only on the final model checkpoint (with score relative to Oracle in parentheses). **Bold** values are within one standard deviation (estimated with 3 runs of Cyclic Cosine) of the best in each column.

| Method | TIC-WIKI-Diff ↓ | | | TIC-STACKOVERFLOW ↓ | | | TIC-STACKE-CAT7 ↓ | | |
|---|---|---|---|---|---|---|---|---|---|
| | Backward | ID | Forward | Backward | ID | Forward | Backward | ID | Forward |
| Cyclic Cosine (std) | 0.033 (0.000) | 0.052 (0.000) | 0.085 (0.000) | 0.041 (0.002) | 0.078 (0.002) | **0.156** (0.003) | 0.045 (0.001) | 0.050 (0.001) | 0.071 (0.000) |
| Cyclic Cosine + AR | 0.033 | 0.054 | 0.087 | **0.032** | **0.077** | 0.159 | 0.035 | **0.044** | 0.068 |
| Cyclic Rsqrt | 0.031 | **0.051** | 0.084 | 0.034 | 0.076 | **0.158** | 0.039 | 0.046 | 0.069 |
| Schedule-Free | 0.035 | 0.055 | 0.087 | 0.038 | 0.079 | 0.160 | 0.045 | 0.050 | 0.072 |
| Replay ($\alpha_t = 1/t$) | 0.038 | 0.063 | 0.091 | 0.075 | 0.121 | 0.191 | 0.036 | 0.052 | 0.072 |
| Replay ($\alpha_t = 1/2$) | 0.032 | 0.055 | 0.086 | 0.055 | 0.094 | 0.170 | 0.038 | 0.049 | 0.070 |
| Replay ($\alpha_t = 1/t$) + AR | 0.039 | 0.063 | 0.092 | 0.066 | 0.119 | 0.193 | **0.031** | 0.050 | 0.072 |
| Replay ($\alpha_t = 1/2$) + AR | 0.033 | 0.057 | 0.088 | 0.047 | 0.096 | 0.176 | **0.032** | 0.046 | 0.071 |
| LwF | 0.033 | 0.053 | 0.085 | 0.037 | **0.075** | **0.155** | 0.044 | 0.048 | 0.070 |
| EWC | **0.030** | **0.051** | **0.083** | **0.033** | **0.077** | 0.162 | 0.035 | **0.043** | **0.067** |

| Method | TIC-CODEDOCS-NUMPY ↓ | | | TIC-CODEDOCS-PYTORCH ↓ | | | Static Evals. ↑ CORE (DCLM) |
|---|---|---|---|---|---|---|---|
| | Backward | ID | Forward | Backward | ID | Forward | |
| Cyclic Cosine (std) | 0.073 (0.004) | 0.096 (0.003) | 0.072 (0.002) | **0.057** (0.002) | **0.025** (0.001) | 0.217 (0.002) | 48.5 (-2.1) (0.4) |
| Cyclic Cosine + AR | 0.054 | 0.074 | 0.062 | 0.084 | 0.052 | 0.228 | 48.5 (-2.1) |
| Cyclic Rsqrt | 0.066 | 0.092 | 0.071 | 0.062 | 0.029 | 0.220 | 49.0 (-1.6) |
| Schedule-Free | 0.069 | 0.100 | 0.080 | 0.084 | 0.051 | 0.236 | **48.8 (-1.8)** |
| Replay ($\alpha_t = 1/t$) | 0.054 | 0.046 | 0.057 | 0.175 | 0.138 | 0.275 | **48.9 (-1.7)** |
| Replay ($\alpha_t = 1/2$) | 0.058 | 0.066 | 0.060 | 0.099 | 0.069 | 0.237 | **49.0 (-1.6)** |
| Replay ($\alpha_t = 1/t$) + AR | 0.040 | **0.045** | 0.057 | 0.197 | 0.169 | 0.277 | **49.0 (-1.6)** |
| Replay ($\alpha_t = 1/2$) + AR | **0.034** | 0.050 | **0.052** | 0.129 | 0.098 | 0.246 | **49.2 (-1.4)** |
| LwF | 0.076 | 0.104 | 0.073 | **0.058** | 0.028 | **0.214** | 48.5 (-2.1) |
| EWC | 0.055 | 0.081 | 0.067 | 0.070 | 0.040 | 0.222 | **48.9 (-1.7)** |

## 6.2 DOWNSTREAM EVALUATIONS

Tab. 3 presents results for several of our downstream evaluations, while Fig. 5 and Appx. D show corresponding evaluation matrices. Here, we observe broadly that methods exhibit trade-offs *across the different evaluations*, showcasing the inherent challenges of performing well on a variety of domains while training on a sequence of generic web-data dumps (in which the coverages of said domains may also vary over time). We summarize the key findings below:

**EWC is the best method for adapting to new knowledge in TIC-WIKI-Diff.** This differs from the results of Garg et al. (2024) and Jin et al. (2022) which found EWC to have little positive impact in their settings. It also differs from TIC-CC-Wiki, where EWC could not match replay in terms of Backward performance. One explanation for this is that by isolating new/changed segments of Wikipedia, TIC-WIKI-Diff places strong pressure on seeing newer data. Indeed, we see in Tab. 8 in Appx. D that measuring performance on the *unchanged* segments returns replay's superiority on Backward, though the improvements are much smaller. To further shed light on this discrepancy, we also see from Fig. 5 (left) that unlike for TIC-CC-WIKI, **peak performance on each TIC-WIKI month is often years *after* that month is seen**, even when no replay is used in Cyclic Cosine. This suggests that the knowledge TIC-WIKI captures each month is also successfully learned from CC dumps that were crawled quite a bit later, which could be due to: (1) delayed alignment between CC's crawls of Wikipedia and TIC-WIKI's more comprehensive coverage of Wikipedia edits; (2) measuring TIC-CC loss on all tokens versus focusing on specific segments and proper nouns in TIC-WIKI to capture only factual knowledge rather than irrelevant nuances (e.g., page formatting).

**Replay of earlier data tends to benefit more on domains expected to evolve slowly.** In Tab. 3, we show both the performance of different methods on TIC-STACKOVERFLOW as well as an average over a subset of seven other large StackExchange sites excluding StackOverflow (TIC-STACKE-CAT7). Earlier CC dumps (i.e., before Feb-2016) are the most useful for TIC-STACKE-MATH leading to improvements from both replay and AR schedules as observed in Fig. 5. In contrast, for TIC-STACKOVERFLOW, there not only exists a larger distribution shift over time, but seeing less old data improves all three summary metrics. Similar to TIC-STACKE, we observe a sharp difference between two evaluations within TIC-CODEDOCS: TIC-CODEDOCS-NUMPY and TIC-CODEDOCS-PYTORCH. As shown in Tab. 3, continual runs involving replay perform better on all metrics for the

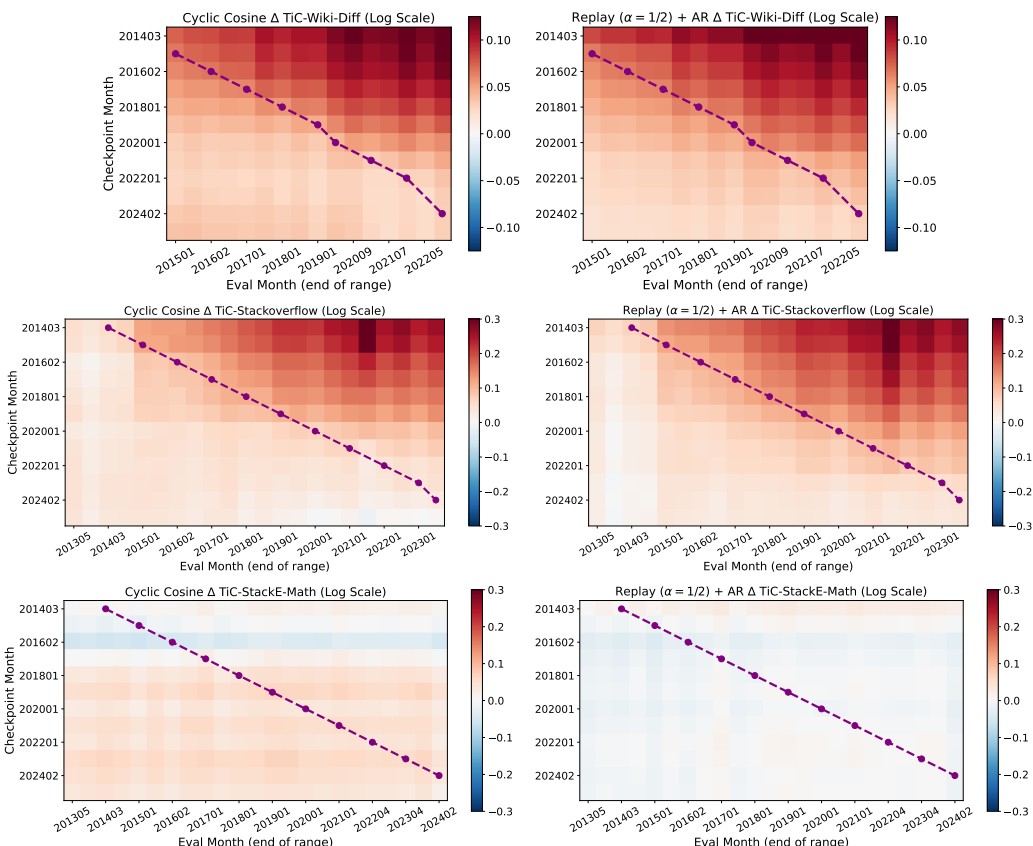

Figure 5: **Replay helps on TIC-STACKE-MATH but hurts on domains where new data is crucial (TIC-STACKOVERFLOW).** We show heatmaps the Cyclic Cosine (left) and Replay ($\alpha_t - 1/2$) + AR methods (right) evaluated on TIC-WIKI, TIC-STACKE-MATH, and TIC-STACKOVERFLOW. The purple dotted lines trace out when the training and evaluation timestamps are closest to one another.

former whereas the opposite is true for latter. This is likely due to NumPy being an older library first released in 1995 compared to PyTorch in 2016. Based on the corresponding heatmaps in Appx. D, it appears that models improve on NumPy in earlier years (i.e., 2013-2016) before forgetting this knowledge when training on the following four to five years. This suggests the bulk of NumPy-related content appeared in earlier years before decreasing, thereby *necessitating* replay for models to retain this knowledge. In contrast for PyTorch, replay harms performance since it shifts more weight to earlier CC dumps, three years of which were before PyTorch even first released.

**Static evaluation of methods with an unbiased initialization matches the Oracle.** Table 3 presents evaluations on the CORE set of tasks from Li et al. (2024a). Initially, we observe that most continual methods perform similarly and a sizable gap to the Oracle remains. Indeed, the initialization trained on the May-2013 already achieves an average of 48.5, the same as the final checkpoint of Cyclic Cosine. Meanwhile, the interventions that mitigate forgetting can somewhat help, closing the remaining gap to the Oracle by 33%. The two possible explanations for the remaining 67% are that the Oracle benefits from: (1) having less restricted access to data throughout its training (i.e., the continual phase of our runs is at fault); (2) being trained from scratch rather than starting from a model biased towards the oldest data (i.e., the initial training on the May-2013 is at fault). To investigate, we run an additional oracle variant which starts from the same May-2013 initialization but trains on an equal mix of the remaining 113 months all at once. This model achieves a performance of only 48.9, below our best continual runs which suggests that (2) is likely more at fault than (1).

### 6.3 EFFICIENCY OF CONTINUAL TRAINING

In this section, we perform a case study on the practical utility of current continual methods by measuring the potential compute savings offered by Replay ($\alpha_t = 1/2$). While the results in previous

| Method | Tokens | TIC-CC | TIC-CC-Wiki | TIC-CC-News | TIC-WIKI-Diff | TIC-WIKI-Unchanged |
|---|---|---|---|---|---|---|
| Replay | 220B | 0.024 | 0.019 | 0.017 | 0.013 | 0.017 |
| Replay + AR | 220B | 0.027 | 0.017 | 0.014 | 0.014 | 0.016 |
| Replay | 330B | 0.011 | 0.009 | 0.010 | 0.001 | 0.006 |
| Replay + AR | 330B | 0.008 | 0.003 | 0.002 | -0.001 | -0.001 |
| Replay | 440B | 0.004 | 0.004 | 0.007 | -0.004 | 0.000 |
| Replay + AR | 440B | **-0.002** | **-0.006** | **-0.005** | **-0.009** | **-0.010** |

| Method | Tokens | TIC-STACKOVERFLOW | TIC-STACKE-Cat7 | TiC-CD-PyTorch | TiC-CD-NumPy |
|---|---|---|---|---|---|
| Replay | 220B | 0.034 | 0.028 | 0.052 | 0.047 |
| Replay + AR | 220B | 0.027 | 0.022 | 0.082 | 0.023 |
| Replay | 330B | 0.014 | 0.020 | 0.003 | 0.035 |
| Replay + AR | 330B | 0.017 | 0.014 | 0.040 | 0.024 |
| Replay | 440B | **0.007** | 0.150 | **-0.025** | **0.018** |
| Replay + AR | 440B | 0.015 | **0.008** | 0.013 | 0.025 |

Table 4: **Replay-based approaches are competitive with re-training Oracles while 62% cheaper.** We scale up two methods that use the ($\alpha_t = 1/2$) version of replay by increasing the monthly token budgets for the continual phase of training. The three scales considered correspond to 220B / 330B / 440B total tokens seen during initialization and continual training. We measure sub-optimality relative to a *series* of Oracle models trained roughly every two years (requiring 1.16T training tokens altogether) and report averages of all Backwards and ID elements of the resulting evaluation matrices.

sections indicate that all continual methods under-perform the Oracle, this Oracle is quite strong for two reasons: (1) for most entries $E_{i,j}$ in our evaluation matrices, it has seen considerably more tokens than model checkpoint $i$; (2) while it is compute matched with continual runs, it would not actually be obtainable until the last month; if one wanted to consistently re-train new models like the Oracle over this 11 year span, the costs would increase linearly with the number of desired updates.

Thus, to more practically measure the cost effectiveness of continual training, we now consider an alternative baseline of a *series* of Oracle models with different data cutoffs. We then measure the sub-optimality of any checkpoint relative to the *series* of Oracles by subtracting the performance of the most recent Oracle (instead of always the Jul-2024 Oracle). Specifically we consider seven cutoff dates roughly spaced two years apart (i.e., May-2013, Jan-2015, Jan-2017, Jan-2019, Jan-2021, Jan-2023, Jul-2024). Each corresponding Oracle is then token matched with the continual checkpoints corresponding to its cutoff date: e.g., the Jan-2019 Oracle is trained on data coming from all months up to Jan-2019 and for the number of tokens seen by a 220B continual run's Jan-2019 checkpoint. In total all seven oracles together require 1.16T tokens. Given that this now costs more than 5× our current continual runs, we also consider increasing the compute budget for continual runs by 1.5× and 2× (by increasing the monthly budget for the continual phase while keeping the initialization the same). These longer runs still cost considerably less than the Oracle series (seeing 330B and 440B tokens respectively), while also being able to update models every month instead of every two years.

**Matching the Oracles with 62% less compute.** In Tab. 4, we report the average of the elements that would have appeared in the Backwards and ID elements of the corresponding matrix. Overall, we observe that scaling up Replay ($\alpha_t = 1/2$) + AR to 440B tokens becomes competitive with the series of Oracles. Despite requiring 62% less compute, it surpasses re-training Oracles on many evaluations (i.e., all TIC-CC and TIC-WIKI subsets) while closing the gap significantly on most others.

## 7 CONCLUSION

We introduce a benchmark for continual LLM pretraining spanning more than a decade of timestamped data. TiC-CommonCrawl (TIC-CC) consists of training and evaluation data spanning more than 100 months. We also introduce new TiC evaluations, TiC-Wikipedia (TIC-WIKI), TiC-StackExchange (TIC-STACKE), and TIC-CODEDOCS. Using these assets, we clearly observe models need to be continually trained to stay up to date but that the ideal update frequency varies based upon the domain, motivating the need for methods that prevent forgetting. To this end, we compared various baseline strategies for continual pretraining, finding that simple cyclic learning rate schedules and data-replay shrink the gap to an *Oracle* that trains on all data. However, completely closing the gap remains an open and challenging problem to be studied by future work on our benchmark.

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

# A DATASET CONSTRUCTION

We build upon the existing pipeline and assets from DataComp-LM (Li et al., 2024a) to build our dataset, only altering steps that rely on global operations across months.

**Initial pool and temporal splitting.** We start with DCLM-Pool (Li et al., 2024a) which contains all CC dumps between May-2013 and December-2022. The only pre-processing that has been done on this pool is to parse the HTML (contained in WARC files of CC) into plaintext for each webpage via the open-source `resiliparse` library (Bevendorff et al., 2018; 2021) [2]. In DCLM-Pool, documents are naturally grouped together into files based upon the CC dump, which is indicated by the file prefix [3]. To split the data by month, we simply group files that share the same prefix. Since DCLM-Pool contains data up to December-2022, we also follow their exact download and extraction scripts to obtain more recent data until July-2024.

**Data preprocessing and tokenization.** Next, we follow DCLM-Baseline's filtering procedure which starts with their implementation of heuristic filters from RefinedWeb. We apply these filters independently on each page with no change. However, we have to modify deduplication that removes nearly identical pages given a similarity threshold. Instead of applying deduplication globally as in DCLM-Baseline, we apply the same deduplication method *only within* each month. Finally, we also skip the final classifier-based filtering in DCLM-Baseline, as their classifier was trained on data that comes from all months, including examples generated by recent LLMs such as GPT-4.

**Data sampling and held-out sets.** DCLM-Pool was partitioned randomly into 10 equally sized "global shards". Within our monthly splits, we also maintain the original global shard assignments. For our training scales, using just one of these global shards within each month is sufficient. Notably though, when we construct evaluation sets such as in (Sec. 4.1), we make sure to sample from a different global shard than the one used for training. This ensures the evaluation data is a sampled from the same distribution as the training data while also being *mostly* held out. Notably, since we do not deduplicate across globals shards or months, there could be overlap between training and eval sets across months. However, we observe from Fig. 6 that potential data leakages are unlikely significantly change relative losses values (compared to the Oracle). For each validation set, we cap the maximum number of tokens to 16.7M which corresponds to 8192 sequences for our context length of 2048. For some months of TIC-CC-WIKI and TIC-CC-NEWS, we end up with less than this amount, but the smallest are 5M and 12M respectively.

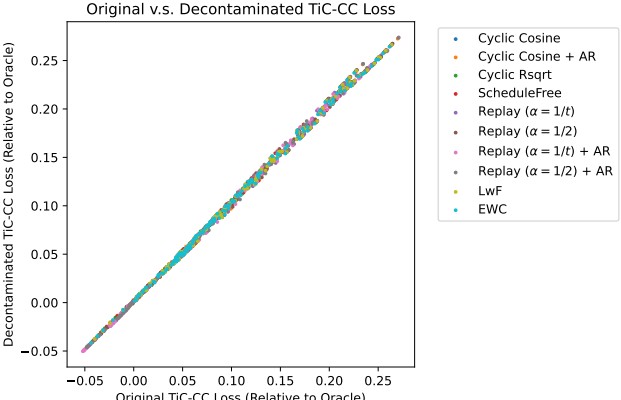

Figure 6: **Findings from TIC-CC are robust to potential data leakages.** We create a decontaminated version of our TIC-CC loss-based evaluation by deduplicating each month's evaluation set using a Bloom Filter pre-populated by the corresponding training set. Overall, across all the methods, checkpoints, and evaluation months we observe strong correlations between using the pre-decontamination (x-axis) and post-decontamination (y-axis) losses (relative to the Oracle).

---

[2]We use `readability` for parsing code documentations in our TIC-CODEDOCS (`https://github.com/mozilla/readability`).

[3]In DCLM-Pool, each file always starts with `CC-MAIN-YYYYMM` where `YYYYMM` indicates the dump month.

# B  DETAILS OF EVALUATIONS

## B.1  TIC-WIKI

We construct TIC-WIKI from Wikipedia and Wikidata which are sister projects from the non-profit Wikimedia Foundation. Wikidata is a structured knowledge graph that stores the structured data of Wikipedia and other sister projects. Data on Wikidata is represented in the form of statements in the form of property-value about an item in the simplest form. For example, "Mount Everest is the highest mountain in the world" is represented as Earth (Q2) (item) → highest point (P610) (property) → Mount Everest (Q513) (value) [4]. The triplet (item, property, value) can also be referred to as (subject, relation, object).

**TemporalWiki dataset generation.** TemporalWiki constructs evaluations from monthly snapshots of English Wikipedia and Wikidata through the following steps:

1. Generate TWiki-Diffsets by identifying changes and additions between consecutive snapshots of Wikipedia. For new articles, the entire article is added to the Diffset while for existing articles, only the changed or new paragraphs are added.

2. Construct TWiki-Probes by processing two consecutive snapshots of Wikidata. Statements are categorized into changed if the property/value has changed or categorized into unchanged otherwise.

3. Align TWiki-Diffsets with Wikidata by ensuring changed statements exist in TWiki-Diffsets and unchanged statements exist in Wikipedia.

4. Heuristic filtering by removing statements where the subject or object is a substring of the other or the object is more than 5 words. Moreover, a single subject is limited to maximum 1% and relation/object is limited to maximum 5% of the total statements.

TIC-WIKI extends TemporalWiki in various ways:

1. Expanding the timespan from four months to a decade (2014-2024), thus capturing a broader spectrum of knowledge evolution.

2. We improve the matching process of Wikipedia and Wikidata dumps, and enhance the robustness of data parser to format changes over time.

### B.1.1  DATA PREPROCESSING

**Wikidata and Wikipedia dumps.** Wikimedia releases regular dumps [5,6], but only retains data for the most recent 4 months. To access historical data, we utilized the Internet Archive [7]. The earliest available dump dates back to November 2014. It is important to note that the archived dumps do not cover every month, with several months missing from the record. In our study, we made use of all available monthly dumps. The filenames of the dumps include the specific date of month that has been collected on, which is typically the 1st or 20th of the month, though this can vary. We include only one dump per month if multiple dumps are available. We check for the first date if not available look for 20th and if neither we start from begining the monthh and check for the first availble date in that month.

**Data cleanup.** We utilize WikiExtractor [8] to clean up the Wikipedia data. This step extracts the main content and removes extraneous and non-essential characters.

**Wikipedia diffsets.** To construct consecutive diffs of Wikipedia, we developed a method comparing snapshots of articles from consecutive dumps. For comparing two snapshots of an article, we first remove extraneous whitespace and standardize formatting by preprocessing the text. This involves removing empty lines, stripping newline characters, and creating a normalized version of each line where punctuation is removed and text is converted to lowercase.

---

[4] https://www.wikidata.org/wiki/Help:About_data
[5] https://dumps.wikimedia.org/wikidatawiki/
[6] https://dumps.wikimedia.org/enwiki/
[7] https://archive.org
[8] https://github.com/attardi/wikiextractor

Afterward, we use a two-level comparison: first at the paragraph level, then at the sentence level for changed paragraphs. We utilize Python's `difflib.SequenceMatcher` to compare the normalized versions of paragraphs and sentences. This hierarchical method, coupled with normalization, captures substantial edits while filtering out minor or stylistic changes.

We extract and store both changed and unchanged content separately. Changed content includes replaced paragraphs with modified sentences and newly inserted paragraphs. Unchanged content preserves paragraphs and sentences that remain identical between versions. New articles are treated as entirely changed content. This approach allows us to focus on meaningful content changes while maintaining the context of unchanged information, providing a comprehensive view of how Wikipedia articles evolve over time. Algorithms 1 and 2 describe the process of constructing Wikipedia diffs and changed/unchanged content.

**Wikidata diffsets.** Next, we extract changed and unchanged Wikidata statements of the form (subject, relation, object) from each consecutive dump. Identical triplets in both dumps are marked as unchanged. Triplets in the new dump not present in the old are categorized as new, with the exception that if a subject entity has more than 10 triplets, the algorithm randomly samples 10 to represent it. When a triplet has the same subject and relation as one in the old dump but a different object and the old and new objects differ only in case (upper/lowercase), the triplet is classified as unchanged; otherwise, it is categorized as new. Triplets from the old dump not found in the new one are implicitly considered removed, but importantly, these are not included in the output sets of changed or unchanged triplets. Throughout this process, the algorithm filters out triplets with overly long object values (more than 5 words) and ensures no duplicates are added. This approach efficiently tracks Wikidata evolution, capturing nuanced changes while managing the volume of data for new entities. Algorithm 3 describes the process of triplet extraction.

**Wikipedia historical dumps.** It is possible to reconstruct each version of Wikiepdia using the large history files Wikipeida provide [9]. There are more than 200 historical dumps of English Wikipedia, each sized more than 2GB. Combined together, these files include all revisions and all pages of Wikipeida.

For Wikidata, Wikimedia does not provide historical diff files as Wikipedia except for the last three months [10]. Wikidata file names are formatted similar to `wikidatawiki-20190101-pages-articles.xml.bz2` and available at URLs similar to `https://dumps.wikimedia.org/wikidatawiki/20240401/`.

Each Wikidata dump is approximately 140GB whereas each Wikipeida dump is less than 22GB. Therefore, it is possible to make a version of Wikipedia that keeps track of all changes which results in 200 files of 2GB. But as far as we know there are no such files for Wikidata.

Using the dumps from `archive.org` has several advantages:

- We are sure that we do not leak information from previous timesteps.
- There exists a Wikidata dump close to each Wikipedia dump to be aligned.
- We can use Wiki-Extractor for filtering and remove Wikipeida editorial discussions.

To illustrate the characteristics of our generated dataset, we present key statistics in the following figures. Figure 7 shows the number of Wikipedia pages with significant changes between consecutive database dumps over time. This graph provides insight into the volume and temporal distribution of our data generation process, highlighting periods of higher and lower content modification as well as distribution of our dumps.

### B.2 TIC-STACKE

#### B.2.1 DATA PREPROCESSING

TIC-STACKE spans data from July 2008 through April 2024. The data was sourced from `archive.org` using the April 2024 dump of StackExchangeEach category in the dump comes with two key

---

[9] `https://dumps.wikimedia.org/enwiki/latest/` file names containing `pages-meta-history`.

[10] `https://dumps.wikimedia.org/wikidatawiki/`

**Algorithm 1** Construct Wikipedia Consecutive Diffs

1: Input: oldSnapshot, newSnapshot
2: Output: changedContent, unchangedContent
3: oldArticles ← ReadArticles(oldSnapshot)
4: newArticles ← ReadArticles(newSnapshot)
5: changedContent ← ∅, unchangedContent← ∅
6: **for** each articleId in newArticles.keys **do**
7:     **if** articleId in oldArticles **then**
8:         oldText ← NormalizeText(oldArticles[articleId].text)
9:         newText ← NormalizeText(newArticles[articleId].text)
10:        changed ← ExtractChangedContent(oldText, newText)
11:        unchanged ← ExtractUnchangedContent(oldText, newText)
12:        Add (articleId, changed) to changedContent
13:        Add (articleId, unchanged) to unchangedContent
14:    **else**
15:        Add (articleId, newArticles[articleId].text) to changedContent
16: **return** changedContent, unchangedContent

---

**Algorithm 2** Extract Changed Content

1: Input: $oldText$, $newText$
2: Output: $changedContent$
3: oldParagraphs ← SplitIntoParagraphs(oldText)
4: newParagraphs ← SplitIntoParagraphs(newText)
5: changedContent ← ∅
6: **for** each (oldPara, newPara) in Zip(oldParagraphs, newParagraphs) **do**
7:     **if** IsDifferent(oldPara, newPara) **then**
8:         oldSentences ← SplitIntoSentences(oldPara)
9:         newSentences ← SplitIntoSentences(newPara)
10:        **for** each (oldSent, newSent) in Zip(oldSentences, newSentences) **do**
11:            **if** IsDifferent(oldSent, newSent) **then**
12:                Add newSent to changedContent
13: **return** changedContent

---

**Algorithm 3** Wikidata Triplet Extraction and Categorization

**Require:** oldDump, newDump
**Ensure:** unchanged, new
  unchanged ← {}
  new ← {}
  newEntities ← {}
  **for all** triplet ∈ newDump **do**
    **if** triplet ∈ oldDump **then**
      Add triplet to unchanged
    **else if** hasSameSubjectPredicate(triplet, oldDump) **then**
      oldObject ← getObject(triplet.subject, triplet.predicate, oldDump)
      **if** equalsIgnoreCase(triplet.object, oldObject) **then**
        Add triplet to unchanged
      **else**
        Add triplet to new
    **else**
      **if** triplet.subject ∉ oldDump **then**
        Add triplet to newEntities[triplet.subject]
      **else**
        Add triplet to new
  sampleNewEntityTriplets(newEntities, new)
  filterLongObjects(unchanged, new)
  removeDuplicates(unchanged, new)
  **return** unchanged, new

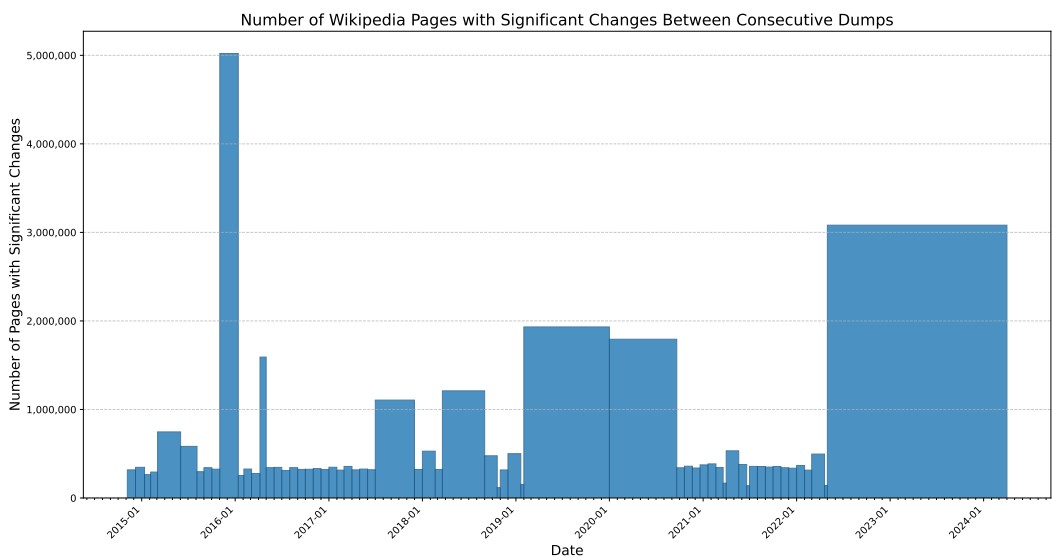

Figure 7: Number of Wikipedia pages with significant Changes between consecutive `archive.org` dumps.

files: `Post.xml` and `PostHistory.xml`. `Post.xml` contains information on how answers and questions relate to each other and includes the latest text for each post entry. `PostHistory.xml` records the changes to each post, whether it is a question or an answer.

To construct our dataset, we first build the graph of question-answer relationships based on the `Post.xml`. We then use `PostHistory.xml` to reconstruct exact snapshots of posts at specific timestamps. This allowed us to capture the state of each post at the end of each month, ensuring our data reflected the actual content available at those points in time.

We construct binary classification tasks from StackExchange content. For each question, we extract two responses: the solution accepted by the original author and an alternative option. Our goal is to create clear distinctions in answer quality, so we implement rigorous selection criteria. Specifically, we requir the accepted solution to have received at least four times the number of upvotes as the alternative. For the alternative, we choose the response with the lowest upvote count that was posted before the accepted answer. This strict filtering, while effective in creating distinct quality differentials, significantly reduced our sample size across most categories. To maintain robust evaluation metrics while preserving data volume, we introduce an additional metric: the average perplexity of accepted answers, calculated without applying the strict upvote ratio filter. This approach allows us to include more samples in our analysis while still capturing meaningful performance trends

We applied this process consistently across all categories of StackExchange, allowing for comprehensive evaluation. In total, we processed 174 out of 182 categories in stackexchange data, of which we focus on `stackoverflow` in this work as well as a group of seven categories: apple, codereview, electronics, english, gaming, math, and worldbuilding. Some categories had insufficient questions in a single month to provide statistically significant results. In such cases, we combined data from consecutive months, ensuring that each time frame contains at least 500 questions.

The full set of sites includes:

3dprinting, academia, ai, android, anime, apple, arduino, astronomy, aviation, avp, beer, bicycles, bioacoustics, bioinformatics, biology, bitcoin, blender, boardgames, bricks, buddhism, cardano, chemistry, chess, chinese, christianity, civicrm, codegolf, codereview, coffee, cogsci, computergraphics, conlang, cooking, craftcms, crafts, crypto, cs, cseducators, cstheory, datascience, dba, devops, diy, drones, drupal, dsp, earthscience, ebooks, economics, electronics, elementaryos, ell, emacs, engineering, english, eosio, esperanto, ethereum, expatriates, expressionengine, fitness, freelancing, french, gamedev, gaming, gardening, genai, genealogy, german, gis, graphicdesign, ham, hardwarerecs, health, hermeneutics, hinduism, history, homebrew, hsm, interpersonal, iot, iota, islam,

italian, japanese, joomla, judaism, korean, langdev, languagelearning, latin, law, lifehacks, linguistics, literature, magento, martialarts, materials, math, matheducators, mathematica, mechanics, meta, moderators, monero, money, movies, music, musicfans, mythology, networkengineering, opendata, opensource, or, outdoors, parenting, patents, pets, philosophy, photo, physics, pm, poker, politics, portuguese, proofassistants, puzzling, quant, quantumcomputing, raspberrypi, retrocomputing, reverseengineering, robotics, rpg, rus, russian, salesforce, scicomp, scifi, security, sharepoint, sitecore, skeptics, softwareengineering, softwarerecs, solana, sound, space, spanish, sports, sqa, stackoverflow, stats, stellar, substrate, sustainability, tex, tezos, tor, travel, tridion, ukrainian, unix, ux, vegetarianism, vi, webapps, webmasters, windowsphone, woodworking, wordpress, workplace, worldbuilding, and writers.

### B.2.2 ANALYSIS OF STACKEXCHANGE DATA

This section presents an analysis of question-answer patterns across the top 20 categories of StackExchange, with a focus on StackOverflow, Mathematics, and English Language & Usage.

**Overall category distribution.** Figure 8 shows the distribution of questions across the top 20 StackExchange categories.

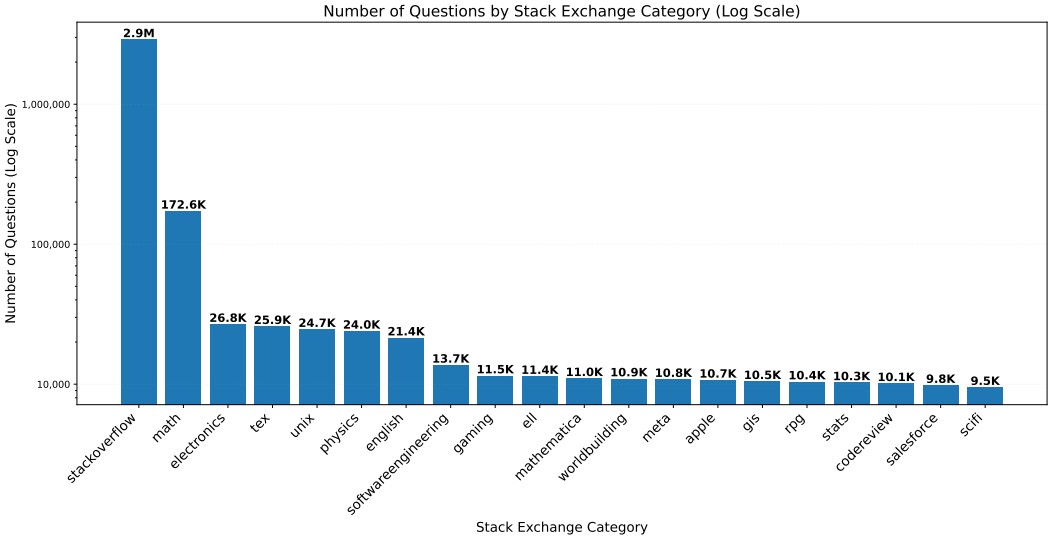

Figure 8: Number of questions by StackExchange category (log scale).

**Temporal trends in question volume.** Figure 9 show the number of questions asked per month for Stack Overflow, Mathematics, and English Language & Usage.

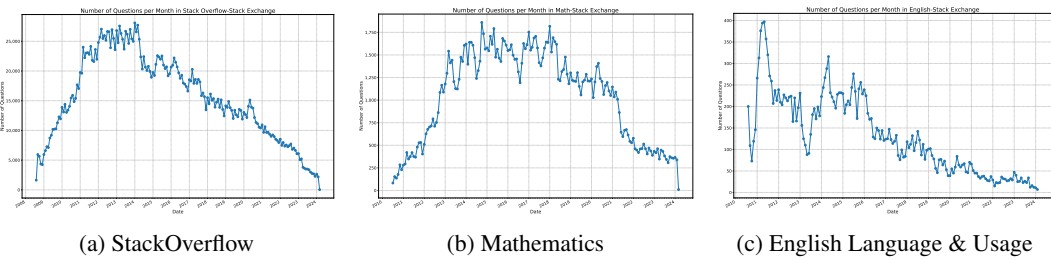

(a) StackOverflow          (b) Mathematics          (c) English Language & Usage

Figure 9: Number of questions per month in StackOverflow, Mathematics and English Language & Usage.

**Question characteristics.** Figure 10 illustrates the distribution of question lengths for StackOverflow, Mathematics, and English Language & Usage.

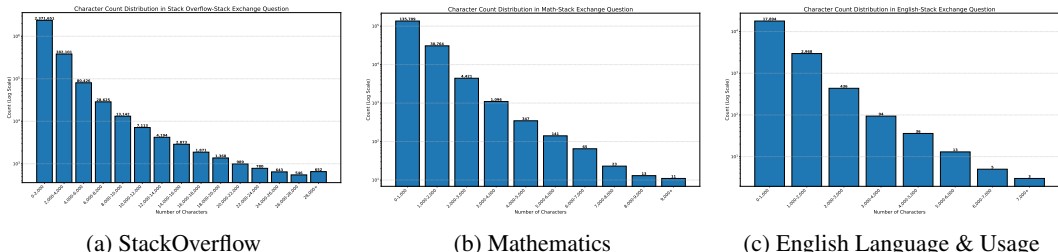

(a) StackOverflow                    (b) Mathematics                    (c) English Language & Usage

Figure 10: Character Count Distribution in StackOverflow, Mathematics and English Language & Usage Questions.

**Answer patterns.** Figure 11 presents the distribution of answer counts per question for StackOverflow, Mathematics, and English Language & Usage.

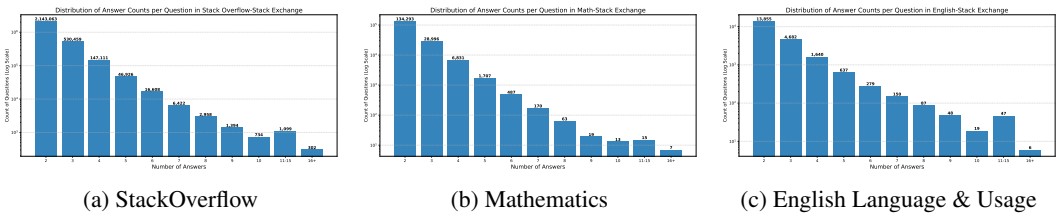

(a) StackOverflow                    (b) Mathematics                    (c) English Language & Usage

Figure 11: Distribution of Answer Counts per Question in Mathematics and English Language & Usage.

## C HYPERPARAMTER TUNING

In general we follow the configurations used in DataComp-LM (Li et al., 2024a) unless further specified. For our Oracle and initialization trained on May-2013, we exactly follow their hyperparameters given that these were also standard pre-training runs from scratch.

For our various continual methods, we do perform additional hyperparameter tuning using the first 10 TiC-CC training sets and held-out validation sets. Following Cha & Cho (2024), we limit the tuning to an early set of months given that it would be impossible for a practitioner to be able to tune based upon data they have not seen far in the future. We discuss the tuning and hyperparamter choices for specific methods in more detail below.

**Cyclic Cosine.** We mainly tuned the maximum learning rate in each cycle, trying values between 1e-3 and 3e-5, as shown in Tab. 5. On our tuning set, the best setting across the board was 1e-4. When carrying out these tuning runs to completion on all 113 timesteps, we do observe an important difference in behavior. While 1e-4 continues to offer the best ID performance and strictly dominates all higher settings, lowering it further can be used to trade-off Backward and ID performance. The smallest fixed max learning rate, 3e-5 results in a similar yet overall worse performance profile to using an an AR meta-schedule. This makes sense given the AR schedule roughly can be considered to decrease the maximum learning rate at a $1/t$ rate; since our setup involves over 100 months, AR schedules set the maximum learning rate very close to the minimum of 3e-5 in most rounds. Overall, we find that learning rates do need to be lowered by at least 30x compared to the the May-2013 initialization (which used 3e-3). This is in contrast to Ibrahim et al. (2024); Gupta et al. (2023) which both suggest re-warming up to a similar learning rate as the initial pre-training or Parmar et al. (2024) who start from the minimum learning rate of the pre-trained model. We suspect this is due to the difference in setup (i.e., these works use only 2 or 3 training rounds of comparable sizes and face distribution shifts related to data quality and language rather than temporal evolution).

Table 5: **Tuning for Cyclic Cosine**

| Max LR | TıC-CC (Tuning Months) | | | (TıC-CC All Months) | | |
|---|---|---|---|---|---|---|
| | Backward | ID | Forward | Backward | ID | Forward |
| 1e-3 | 0.103 | 0.086 | 0.118 | 0.197 | 0.083 | 0.209 |
| 3e-4 | 0.019 | 0.016 | 0.051 | 0.125 | 0.041 | 0.178 |
| 1e-4 | **0.002** | **0.005** | **0.039** | 0.072 | **0.027** | **0.161** |
| 5e-5 | **0.002** | 0.006 | **0.039** | 0.062 | 0.034 | 0.163 |
| 3e-5 | 0.004 | 0.009 | 0.040 | 0.060 | 0.042 | 0.165 |
| AR Schedule | **0.002** | 0.008 | 0.043 | **0.058** | 0.040 | 0.166 |

**Rsqrt.** We tuned both the maximum learning rate within the same range as Cyclic Cosine as well as the cooldown length, choosing between 50 and 400. Our final run continued to use 1e-4 for the maximum learning rate and 400 for the cooldown, though there did not appear to be much difference when compared to smaller values such as 200 or 100 on the tuning months.

**Schedule-Free.** We continued to use warmup but given that Schedule-Free makes more drastic changes to optimization (i.e. using a different optimizer versus simply a different learning rate schedule), we re-tuned both the learning rate and weight decay. Interestingly, 1e-4 as the maximum learning rate continued to work best for us, though we found it helped slightly to drop the weight decay from 0.033 to 0.01.

Table 6: **Tuning for Schedule-Free**

| Max LR | WD | TıC-CC (Tuning Months) | | |
|---|---|---|---|---|
| | | Backward | ID | Forward |
| 1e-3 | 0.033 | 0.1025 | 0.0856 | 0.1178 |
| 5e-4 | 0.033 | 0.0448 | 0.0373 | 0.0713 |
| 3e-4 | 0.033 | 0.0206 | 0.0183 | 0.0532 |
| 5e-5 | 0.033 | 0.0053 | 0.0105 | 0.0406 |
| 1e-4 | 0.067 | 0.0049 | 0.0080 | 0.0406 |
| 1e-4 | 0.033 | 0.0044 | 0.0077 | 0.0404 |
| 1e-4 | 0.010 | **0.0042** | **0.0075** | **0.0403** |
| 1e-4 | 0.005 | **0.0042** | **0.0075** | **0.0403** |
| 1e-4 | 0.0001 | 0.0044 | 0.0077 | 0.0404 |

**LwF.** Following the original paper (Li & Hoiem, 2018), we used a temperature parameter of $T = 2$. We mainly tuned the regularization weight $\lambda$ trying values between 0.1 and 1.0 and settling upon 0.3. However, overall we found using LwF either resulted in little difference (when using a small $\lambda$) or started to decrease all metrics (when using a larger $\lambda$).

**EWC.** We fixed the number of iterations used to estimate the Fisher matrix to 100 and similar to LwF, we focused on tuning the weight given to the EWC regularization term. Overall, we found that fairly high values were needed to overcome the small values in the approximate Fisher matrix (coming from small second order moment terms). We found that $\lambda = 10^7$ performed best when tuning between $10^1$ and $10^9$, as shown in Tab. 7. The only other setting we tried that is not strictly dominated by this choice was $\lambda = 10^6$, which resulted in slightly better ID performance but significantly worse backward transfer.

Table 7: **Tuning $\lambda$ for EWC**

| $\lambda$ | TIC-CC (Tuning Months) | | |
|---|---|---|---|
| | Backward | ID | Forward |
| $10^0$ | 0.0025 | 0.0050 | 0.0394 |
| $10^1$ | 0.0025 | 0.0050 | 0.0394 |
| $10^4$ | 0.0025 | 0.0050 | 0.0394 |
| $10^5$ | 0.0025 | 0.0049 | 0.0394 |
| $10^6$ | 0.0021 | **0.0047** | 0.0391 |
| $10^7$ | **0.0013** | 0.0050 | **0.0389** |
| $10^8$ | 0.0107 | 0.0178 | 0.0462 |
| $10^9$ | 0.0286 | 0.0400 | 0.0586 |

# D EXTENDED RESULTS

## D.1 TIC-COMMONCRAWL (TIC-CC) VALIDATION SETS

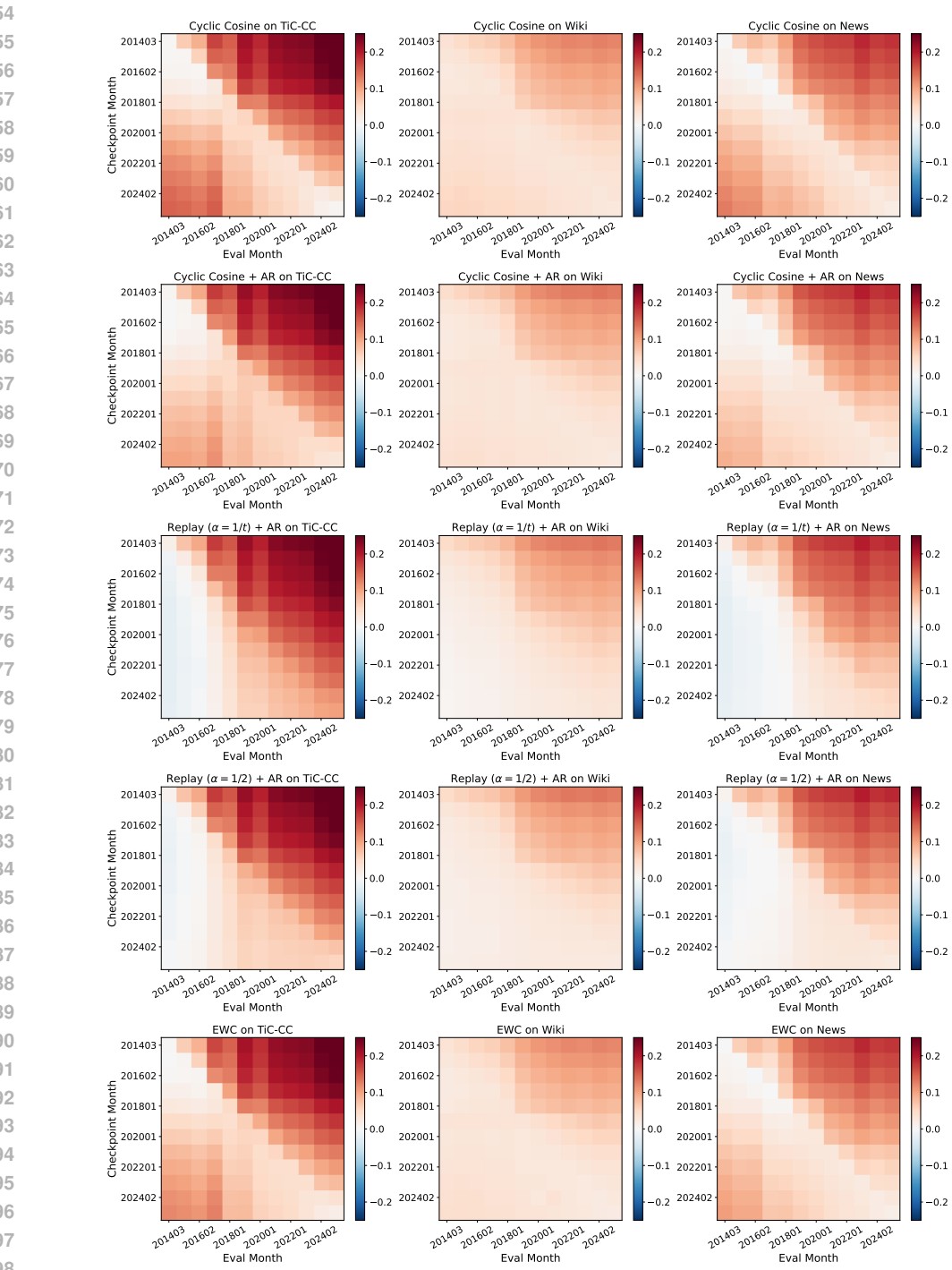

Figure 12: Evaluation matrix heatmaps for selected methods on our TIC-CC evaluations.

## D.2   TIC-WIKIPEDIA (TIC-WIKI)

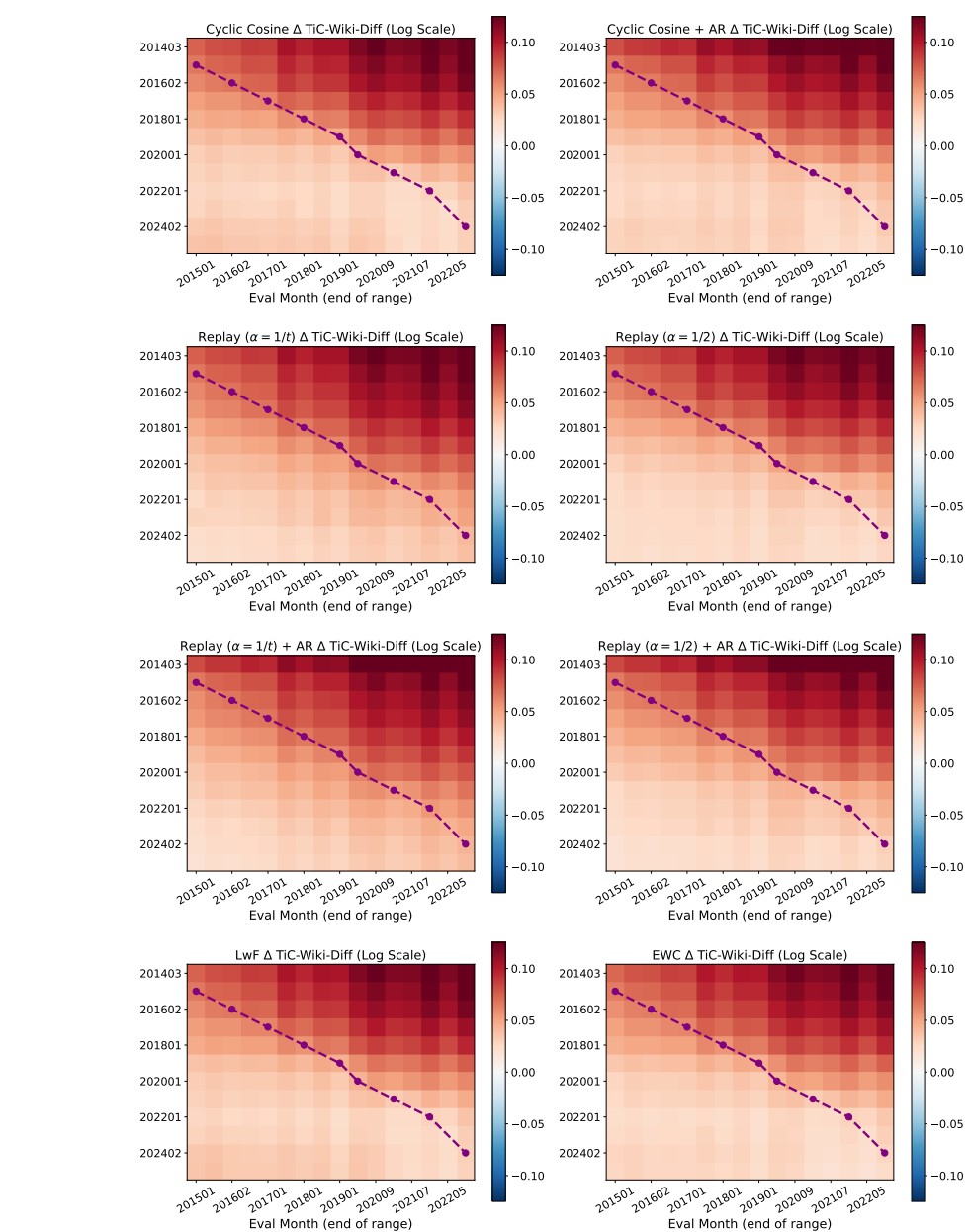

Figure 13: Evaluation matrix heatmaps for various methods on TIC-WIKI.

Table 8: **Comparing TIC-WIKI-Diff versus TIC-WIKI-Unchanged**

| Method | TIC-WIKI-Diff ↓ | | | TIC-WIKI-Unchanged↓ | | |
|---|---|---|---|---|---|---|
| | Backward | ID | Forward | Backward | ID | Forward |
| Cyclic Cosine (std) | 0.033 (0.000) | 0.052 (0.000) | 0.085 (0.000) | 0.039 (0.000) | 0.052 (0.000) | 0.072 (0.000) |
| Cyclic Cosine + AR | 0.033 | 0.054 | 0.087 | 0.035 | 0.051 | 0.074 |
| Cyclic Rsqrt | 0.031 | **0.051** | 0.084 | 0.035 | 0.050 | 0.070 |
| Schedule-Free | 0.035 | 0.055 | 0.087 | 0.040 | 0.055 | 0.074 |
| Replay ($\alpha_t = 1/t$) | 0.038 | 0.063 | 0.091 | 0.035 | 0.056 | 0.074 |
| Replay ($\alpha_t = 1/2$) | 0.032 | 0.055 | 0.086 | 0.034 | 0.053 | 0.072 |
| Replay ($\alpha_t = 1/t$) + AR | 0.039 | 0.063 | 0.092 | 0.034 | 0.055 | 0.077 |
| Replay ($\alpha_t = 1/2$) + AR | 0.033 | 0.057 | 0.088 | **0.033** | 0.052 | 0.074 |
| LwF | 0.033 | 0.053 | 0.085 | 0.039 | 0.053 | 0.072 |
| EWC | **0.030** | **0.051** | **0.083** | 0.034 | **0.050** | **0.069** |

## D.3  TiC-STACKEXCHANGE (TiC-STACKE)

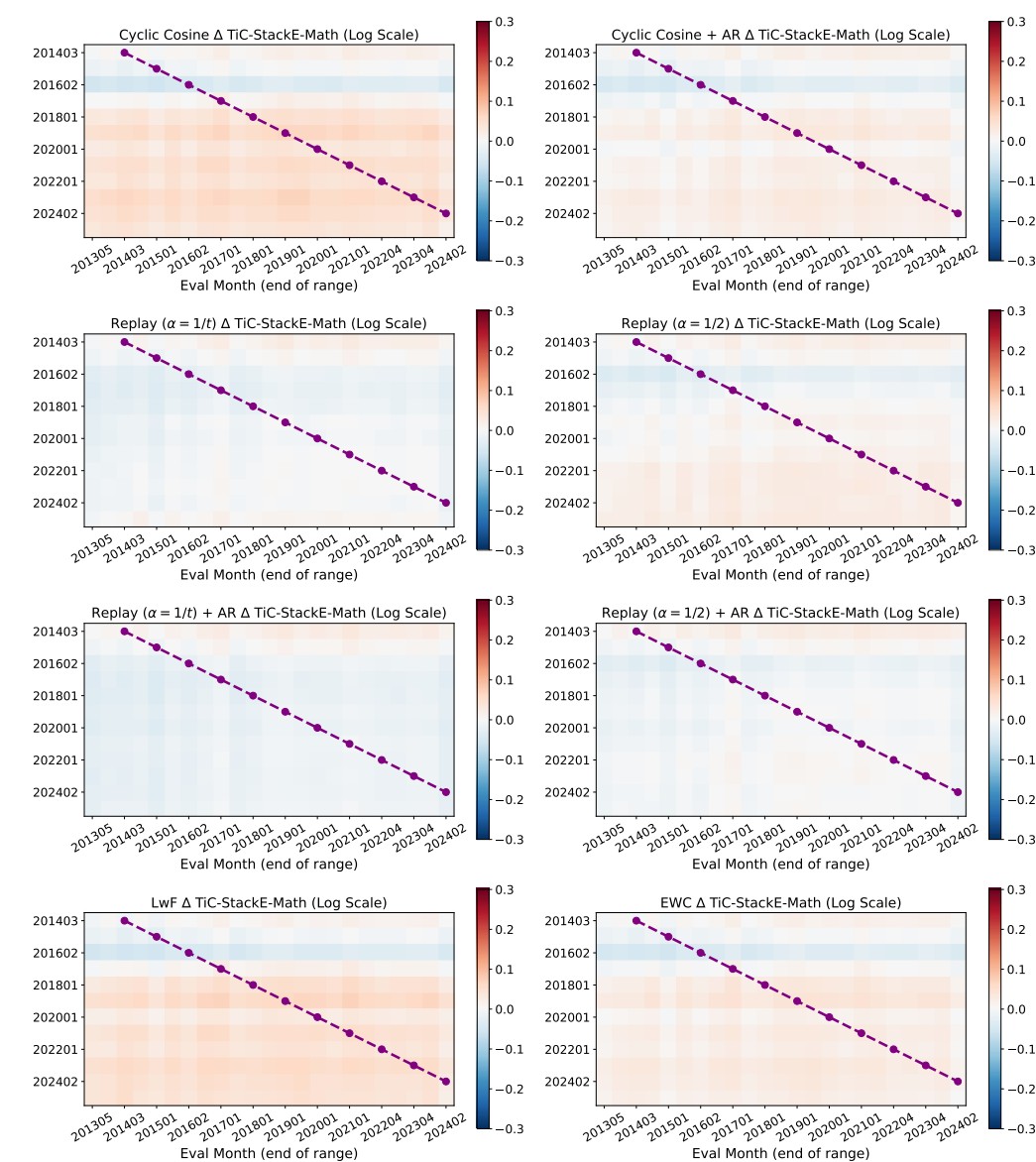

Figure 14: Evaluation matrix heatmaps for various methods on the Math site of TiC-STACKE.

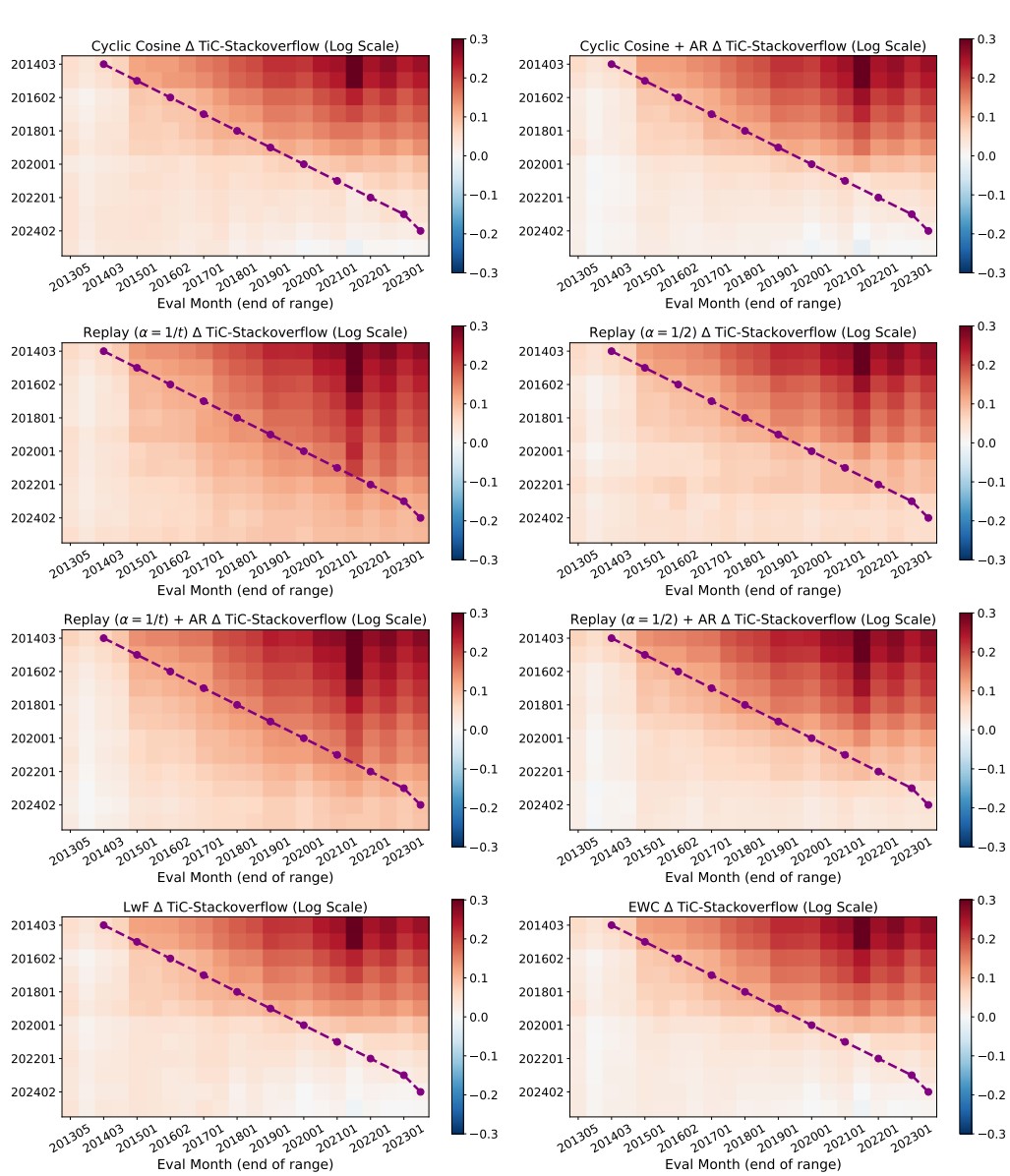

Figure 15: Heatmaps for various methods on the StackOverflow site of TIC-STACKE.

Table 9: Average over an extended set of TIC-STACKE evaluations that we refer to as TIC-STACKE-CAT7. This includes the following sites: apple, codereview, electronics, english, gaming, math, and worldbuilding. Overall, we find that a combination of replay and AR meta-schedules does the most to reduce forgetting while EWC performs best on ID and Forward evaluations.

| Method | TIC-STACKE-CAT7↓ | | |
|---|---|---|---|
| | Backward | ID | Forward |
| Cyclic Cosine (std) | 0.045 (0.001) | 0.050 (0.001) | 0.071 (0.000) |
| Cyclic Cosine + AR | 0.035 | **0.044** | 0.068 |
| Replay ($\alpha_t = 1/t$) | 0.036 | 0.052 | 0.072 |
| Replay ($\alpha_t = 1/2$) | 0.038 | 0.049 | 0.070 |
| Replay ($\alpha_t = 1/t$) + AR | **0.031** | 0.050 | 0.072 |
| Replay ($\alpha_t = 1/2$) + AR | **0.032** | 0.046 | 0.071 |
| LwF | 0.044 | 0.048 | 0.070 |
| EWC | 0.035 | **0.043** | **0.067** |

## D.4    TIC-CODEDOCS

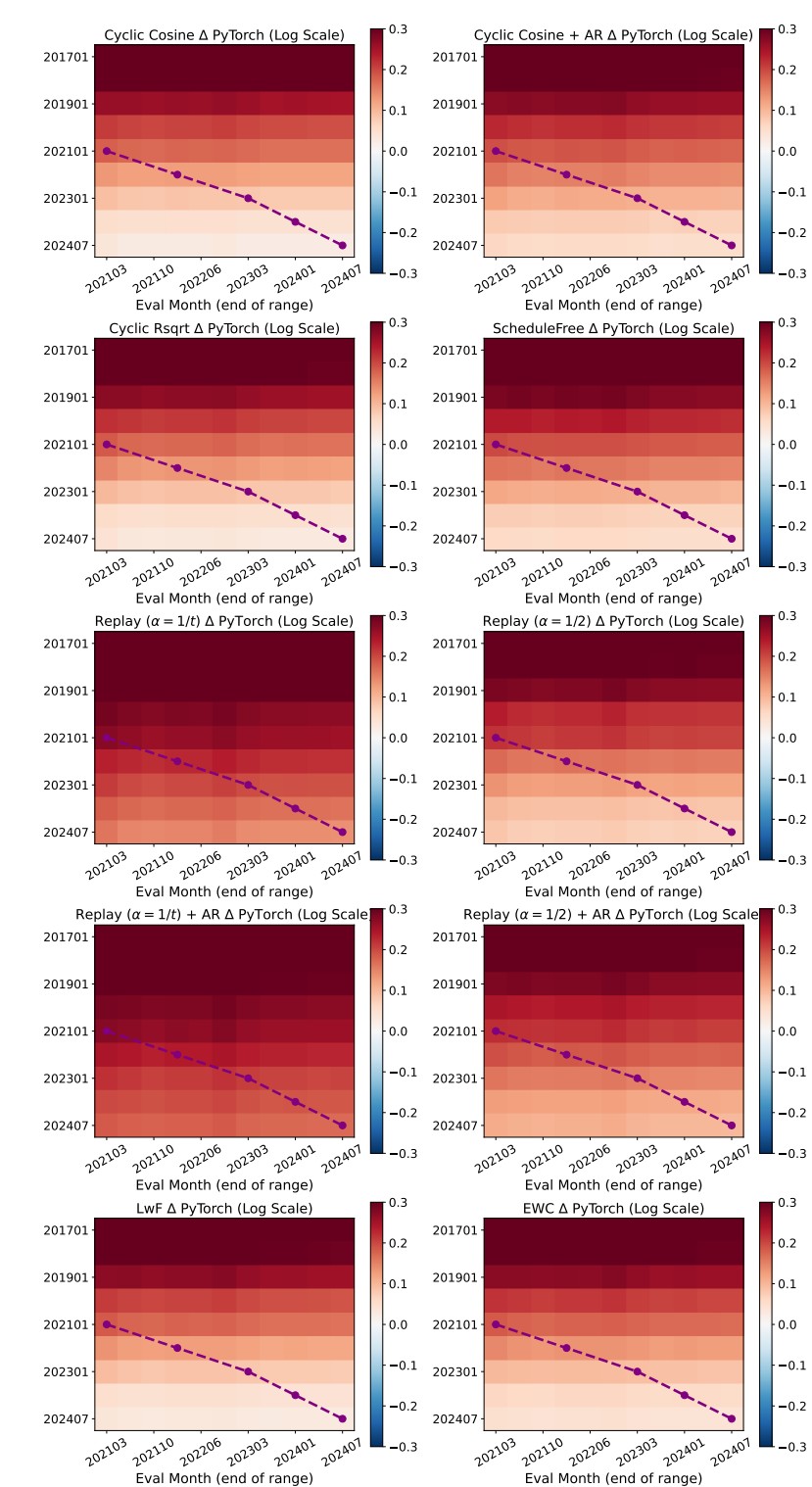

Figure 16: Heatmaps for various methods on TIC-CODEDOCS-PYTORCH

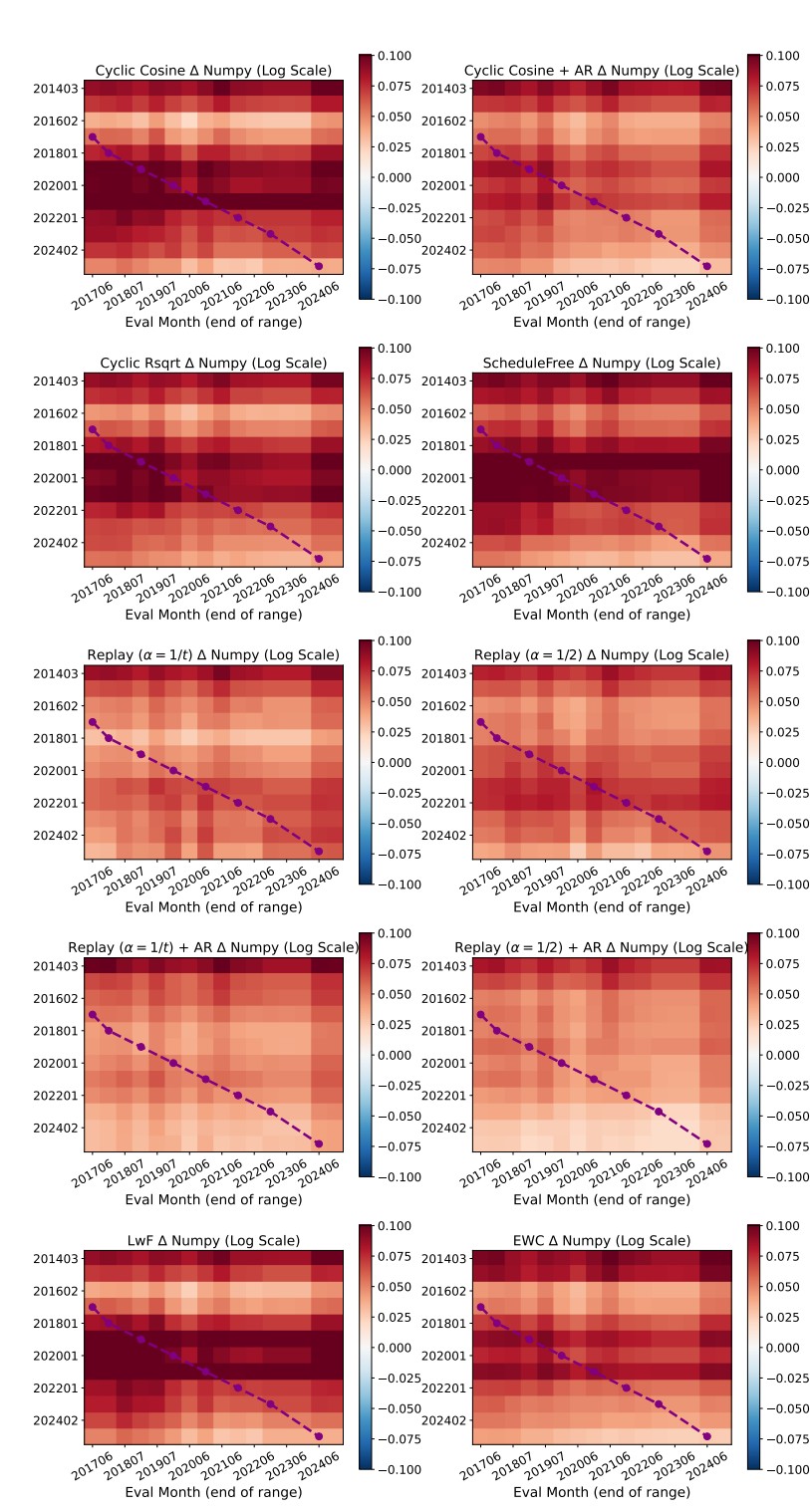

Figure 17: Heatmaps for various methods on TIC-CODEDOCS-NUMPY

# E  EXTENDED RELATED WORK

**Temporal knowledge evaluations.** Language models are expected to have an understanding of time to answer questions about specific time periods and generally reason about time. Various benchmarks have been proposed to evaluate temporal knowledge of LLMs. TemporalWiki (Jang et al., 2022a) evaluates the capability of models to update factual knowledge. TemporalWiki is constructed from the difference between four consecutive snapshots of Wikipedia and Wikidata. Our TIC-WIKI evaluation expands and improves on TemporalWiki in various ways (see Appx. B.1). StreamingQA (Liška et al., 2022) consists of human written and generated questions from 14 years of news articles. The evaluation is either open-book where a model receives a collection of news articles that contain the answer, or closed-book where the model is first fine-tuned on the training set containing the documents and then tested. We evaluate our TiC checkpoints on StreamingQA both in open/closed-book setups and find that there is high ambiguity in the questions that evaluates reasoning more than temporal knowledge understanding. TempEL (Zaporojets et al., 2022) evaluates entity linking performance across 10 yearly snapshots of Wikipedia. Entity linking is the task of mapping anchor mentions to target entities that describe them in a knowledge base. In comparison, our TIC-WIKI evaluates general language and knowledge understanding. TempLAMA (Dhingra et al., 2022) constructs an evaluation for factual queries from Wikidata. They focus on temporally sensitive knowledge with known start and end dates in a specific Wikidata snapshot. Notably, they propose TempoT5 to jointly model text and timestamp which allows a language model to answer temporal questions that change over time such "Who is the president". EvolvingQA (Kim et al., 2024) is also a benchmark for training and evaluating on Wikipedia over time where a LLM automatically generates question-answers from 6 months of articles in 2023. We avoid using any LLMs for generating our evaluations to prevent transfer of bias. TIQ (Jia et al., 2024) benchmark consists of 10k questions-answers based on significant events for the years 1801–2025.

**Temporal generalization.** Beyond understanding the past, LLMs need to be prepared for the future. Li et al. (2024b) observes performance deterioration of public LLMs on Wikipedia, news, code, and arXiv papers after their training data cutoff date. They particularly use compression rate achieved by treating an LLM as a general input compressor using arithmetic coding (Delétang et al., 2024). Our comprehensive evaluations on CommonCrawl, Wikipedia, news articles, StackExchange, and code evaluations verifies their results and more comprehensively shows that the rate of deterioration is domain-specific. DyKnow (Mousavi et al., 2024) evaluations also reaffirm that LLMs private and open-source LLMs have outdated knowledge by asking them questions constructed using Wikidata. They also observe LLMs output inconsistent answers in response to prompt variations and current knowledge editing methods do not reduce outdatedness. TAQA (Zhao et al., 2024) further demonstrate that pretrained LLMs mostly answer questions using knowledge from years before their pretraining cutoff. They construct question/answers from Wikipedia for years 2000–2023 and propose three methods to improve the temporal alignment of models. Similar observations have been made in RealTimeQA (Kasai et al., 2024) and TempUN (Beniwal et al., 2024). These works further solidify the need for continuously updating models with continual pretraining.

**Temporal understanding.** General temporal understanding involves reasoning based on the relation between existing knowledge. Test of Time (Fatemi et al., 2024) benchmark evaluates temporal reasoning, logic, and arithmetics by constructing a synthetic dataset. Their goal is to reduce the chance of factual inconsistency in the evaluation using synthetic data. Our benchmark is designed to be fully realistic based on real data and timestamps to understand the challenges of large-scale continual pretraining in practice. Gurnee & Tegmark (2024) find that LLMs learn a representation of space and time with individual neurons that encode spatial and temporal coordinates. They construct datasets of named entities and find that linear probing LLMs performs well on predicting spatial and temporal coordinates. Nylund et al. (2024) proposed time vectors that specify a direction in the model's weight space that improve performance on text from a specific time period.

**Temporal domain-specific evaluations.** We can further analyze the temporal understanding of a model based on the performance on specific domains with varying rates of change. Luu et al. (2022) studied temporal misalignment such as quantifying temporal degradation of domain-specific finetuning in four domains: social media, science, news, and food reviews. They observed significant temporal degradation in domains such as news, social media, and science but less in food reviews. Gururangan et al. (2020) studied domain-adaptive pretraining and task-adaptive pretraining on unlabeled data for four domains in science, news, and reviews. They observe domain/task-adaptive

pretraining improves performance on the new domain but do not evaluate forgetting on previous domains. Agarwal & Nenkova (2022) studies the temporal model deterioration on future evaluations. They find that the deterioration is task-dependent and domain-adaptive pretraining does not help hypothesizing that limited pretraining data is detrimental in continual pretraining. Jin et al. (2022) domain-incremental pretraining for four scientific domains as well as temporal pretraining on social media over 6 years. They focus on the impact on downstream performance after fine-tuning. They observe distillation-based approaches are the most effective in retaining dowstream performance for tasks related to earlier domains. Overall, the gap between different continual learning methods remained small that can be due to the small scale of pretraining. In comparison, our TIC-CC training is simulating large-scale pretraining.

**Domain/task-continual learning for LLMs.** In domain/task continual learning, the model is presented with a sequence of tasks with predefined labels (Hsu et al., 2018; Van de Ven & Tolias, 2019; Zhou et al., 2023). Each task comes with its training and test sets. In contrast with continual pretraining, the model needs to support a growing set of labels while compared with temporal continual learning, the order of tasks are often arbitrary (e.g., Split-CIFAR, Perm-MNIST). Prominent methods in this domain are regularization-based methods (Kirkpatrick et al., 2017; Mirzadeh et al., 2020a;b; Farajtabar et al., 2020), replay-based methods that often perform superior (Lomonaco et al., 2022; Balaji et al., 2020; Prabhu et al., 2020), and architecture-based methods that adapt the model over time (Schwarz et al., 2018; Rusu et al., 2016). Continual learning for language models has also been dominated by domain/task continual works. Jin et al. (2022) proposed benchmarks for continually training models on a sequence of research paper domains as well as chronologically-ordered tweet streams. Razdaibiedina et al. (2023) proposed learning a new soft prompt for each task and pass soft prompts for all seen tasks to the model which provides adaptability while preventing catastrophic forgetting. Luo et al. (2023) studied continual learning for instruction tuning and observed catastrophic forgetting, especially for larger models. Mehta et al. (2023) showed that generic pretraining implicitly reduces catastrophic forgetting during task incremental finetuning.

**Continual pretraining of LLMs.** Recent work have studied continual pretraining of foundation models at large-scale. TiC-CLIP (Garg et al., 2024) proposed a benchmark of training and evaluation of image-text foundation models and demonstrated the deterioration of existing foundation models on new data. Lazaridou et al. (2021) studied time-stratified language pretraining on WMT, news, and arXiv up to 2019 and observed the models become outdated quickly on news data that holds even for models of various sizes. They study dynamic evaluation as a continual pretraining method that trains on a stream of chronologically ordered documents and observed that models can be updated. However, they did not explore the impact on forgetting and scalability of the method to more generic pretraining data over years. Jang et al. (2022b) proposed continual knowledge learning as a new problem and suggested that parameter expansion is necessary to retain and learn knowledge. They focus on one-step continual pretraining where models are pretrained on C4/Wikipedia data up to 2020 and then trained once more on recent news articles. They find adapter methods perform better than regularization and replay methods. Adapter methods are not directly applicable in our multi-year continual pretraining setup where we train in more than 100 steps on large-scale data. Gupta et al. (2023) proposed simple recipes for continual pretraining of LLMs such as utilizing cyclical learning rate schedules with warmup and ablated on hyperparameters such as warmup duration when continuing the pretraining on a fixed pair of pretraining datasets.

**Time-aware training.** Orthogonal to continual pretraining, one can modify the training or fine-tuning of a model to include explicit information about time. TempLAMA (Dhingra et al., 2022) proposed prepending a time prefix to each example during training which gives the model the flexibility to respond to time-sensitive questions. They train models on news articles where the time can be reliably extracted. Drinkall et al. (2024) proposed training a series of models with sequential data cutoffs dates to avoid data contamination with benchmark and private data. The observe no difference across time on static downstream evaluations when training models on news and Wikipedia

**Factual editing and retrieval augmented generation (RAG).** Another set of works aims to address the staleness of pretrained LLMs without further standard pretraining. One approach is to surgically edit the facts a model "knows" by identifying and updating the relevant weights within a model (Mitchell et al., 2022a). Another is to store edits in an explicit memory and learn to reason over them (Mitchell et al., 2022b). Retrieval augmented generation (RAG) pairs an LLM with new data sources to retrieve the most relevant document for a query. Generally, continual pretraining and RAG are orthogonal approaches to generate up to date responses. RAG methods increase the

cost at inference time without changing the model while continual pretraining is the opposite. Fresh-LLMs (Vu et al., 2024) proposes a QA benchmark and argues that fast-changing knowledge requires a retrieval-based solution compared with slow-changing knowledge. Continual pretraining can be crucial in reducing the cost of RAG by utilizing retrieval only on knowledge that changes faster than the rate of continual pretraining.

## F  FUTURE WORK

**Tokenizer.** As the data changes over the years, new words appear in the language that would benefit from temporal adaptation of the tokenizer (Zheng et al., 2024). In this work, we fixed the tokenizer and did not change it across models. One important challenge that changing the tokenizer introduces is that the perplexity of models with different vocabularies will not be directly comparable. Future work would need to either focus on non-perplexity evaluations (Delétang et al., 2024) or normalize perplexity by a mapping between vocabularies of a checkpoint to the reference oracle model.

**Joint training of text and timestamp.** TIC-CC training data has monthly timestamp corresponding to the crawl time that could be passed as context to the LLM during training and evaluation. TempoT5 (Dhingra et al., 2022) and TempoBERT (Rosin et al., 2022) explored temporal language modeling for example by prefixing the input with "Year: " which helps resolve ambiguity in knowledge that has time-dependent answers such as "Who is the president".

