# OpenReview forum: "TiC-LM: A Multi-Year Benchmark for Continual Pretraining of Language Models"
_ICLR.cc/2025/Conference — Submitted to ICLR 2025_

### Official Review · Reviewer_YDQj · 2024-10-20

**Soundness:** 2
**Presentation:** 3
**Contribution:** 3
**Rating:** 6
**Confidence:** 4

**Summary:**

This paper curates a multi-year benchmark for evaluating continual training methods based off Common Crawl (CC). Moreover, it also designs various dynamic evaluations based on CC, wikipedia, StackExchange and codes, to evaluate forgetting and forward transferring. The paper contains various experiments on the top of benchmarks to measure the effect of different scheduler, regularization, and data replay methods. The main conclusion includes LLM needs to be updated continually to stay up-to-date, and data replay mitigate the issue of forgetting but not eliminate it.

**Strengths:**

1. This paper offers significant contribution to the community by presenting the largest continual learning benchmark for the community to evaluate and compare CL techniques.
2. The experiment design is fairly comprehensive, offer a baseline study for CL technique evaluation.
3. I find the paper is clear and well-displayed. The selected metric is clear and powerful; visualization is informative easy to understand. The paper is well-structured overall.
4. The empirical insights drawn from the CL evaluation experiments are original. I believe they are beneficial to researchers working in the space of LLM continual learning.

**Weaknesses:**

1. Although the presentation is comprehensive dataset is valuable, the CL dataset curation and downstream evaluation dataset curation are extension of existent approaches. TiC-CC is based on the  DCLM pipelines, TiC-Code, TiC-Wiki and TiC-StackExchange are curated using fairly standard approaches.

2. The metrics used in the paper only capture the training loss (perplexity), and forgetting based on the perplexity loss. Other dimensions of forgetting is not measured. The relation between perplexity loss and model behavior (e.g., knowledge forgetting) is not studied. For example, data replay helps reduce the gap with Oracle in Figure 4 on wiki, but it doesn't help on wiki-diff in Table 3.

**Questions:**

1. In the bottom panel of Figure 4, all three figures exhibits that later checkpoints have smaller gap with Oracle on earlier evaluation month. For example, the checkpoint of 202402 records gap ~0 on 2014 but increases gradually to 202402 on TiC-CC and TiC-New. The trend is not as clear on Wiki but exists. Shouldn't all figures follow similar pattern with the top left figure despite with or without replay? How do you explain the observation?

2. Data replay doesn't seem help on most of the downstream evaluations and even harm the Backward performance in some cases (Tic-Wiki-Diff, StackOverflow, CodeDocs-Pytorch), but it shows clear benefits in the TiC-CC evaluations. This appears to be an anomaly, doesn't it?

---

> ### Author Response · Authors · 2024-11-24
> **Reply to Reviewer YDQj [1/2]**
>
> Thank you for your time and thoughtful feedback! We hope to address your concerns and questions below.
>
> **Novelty of dataset/evaluations.** Generally, we see the scale and scope of the TiC-LM as its most important contribution. To our knowledge, it is the first continual learning benchmark that captures two key aspects of LLM pre-training : (1) the time-evolving training data is generic web-data rather than limited domains such as Wikipedia/Tweets/News; (2) the evaluations span a variety of domains, reflecting the general goal of LLMs to be good across many possible contexts.
>
> Based upon (1), we intentionally designed the training data to build upon the preprocessing of existing strong pre-training datasets (i.e., DCLM, RefinedWeb) so as to provide meaningful insights on realistic pre-training distributions, though alternative data-centric interventions would certainly be interesting as future work!
>
> For evaluations, we see (2) as providing significant value on its own, especially as we showcase that different methods (e.g. replay) are more/less effective depending on domain. On top of this, we would like to highlight some technical contributions of our evaluations, which we described in the Appendix but will do better to highlight in future revisions:
>
> * TiC-Wiki dramatically increases the evaluation timespan of the original TemporalWiki from 4 months to 10 years. Also, as discussed in more detail in App. B.1, we used own parsing to handle format changes and maintain temporal consistency across Wikipedia/Wikidata dumps. At a high-level, the approach used by TemporalWiki relies on hardcoded offsets to extract Knowledge Graph triples of Wikidata and requires web API requests to match Wikipedia pages with Wikidata IDs. Our approach, amongst other optimizations, instead systematically processes dumps of both Wikidata and Wikipedia to improve robustness and efficiency.
> * TiC-StackExchange is a new evaluation we created which spans a wide time-range (>10 years) and diversity of topics/domains via the many different StackExchange sites. From a technical point of view, it involved careful reconstruction of historical post states which we share in our scripts.
> * TiC-CodeDocs is the first evaluation based on API documentation evolution, requiring version-aware parsers to process 32 major releases of NumPy and PyTorch while handling complex format changes.
>
> Overall, we plan to open-source our data generation scripts that will also allow for generating variations of evaluation sets (e.g. different time granularities) as well as the unified framework that we used to run all evaluations together.
>
> **Evaluation metrics.** We primarily report perplexity results since our work focuses on base model evaluation without instruction tuning. Practically speaking, perplexity can be evaluated on any domain or data slice without any need for specialized formatting and also allows meaningful comparisons across domains (whereas QA-based evaluations would highly depend on the inherent difficulty of the QA creation method/format). In general, at our compute scales, the variance of perplexity metrics in our evaluations was also small enough to allow us to observe clearer differences between methods.
>
> Regarding the contrast between TiC-CC-Wiki and TiC-Wiki-Diff, we believe this occurs because of the different ways they are constructed. The former is based on all tokens of any Wikipedia page in TiC-CC (exactly as they appear) whereas the latter differs in that it: (a) isolates specifically the most recently changed page segments; (b) focuses on factual information (by measuring perplexity only on proper nouns) (c) more comprehensively covers Wikipedia instead of whatever CC happened to crawl (which may over-represent some topics). Regarding factor (a), since the submission we also measured performance on TiC-Wiki-Unchanged (see Tab. 8 in App. D.2), which is based upon unchanged segments across Wikipedia dumps. On this evaluation, we observe that Replay + AR schedules return to offering the best backward transfer, though the gap is much smaller than on TiC-CC-Wiki. We speculate this could perhaps be due to factors (b) and (c). In our updated manuscript, we have added this discussion to Sec 6.2.

---

> ### Author Response · Authors · 2024-11-24
> **Reply to Reviewer YDQj [2/2]**
>
> **Clarification on Figure 4.** We believe this happens because when using replay, the earlier months will be seen for more tokens throughout the continual run (as they are replayed the most amount of times). This is also why we see some trade-offs between ID and backwards performance when using replay or not in Table 2. In our updated manuscript, we have mentioned this more explicitly in Sec 6.1.
>
> **Impacts of replay.**  We’d like to clarify that one of our takeaways is that on some downstream evaluations, replay is actually crucial such as TiC-StackE-Math and TiC-CodeDocs-NumPy. For the evaluations that you mentioned, they either directly try to isolate the most recent information (TiC-Wiki-Diff) or cover domains/topics that are more likely to evolve quickly (StackOverflow, Pytorch). In general, we see the varying impacts of replay as an important finding from our study, one that results from the realities of pre-training on generic web-data (as opposed to the single-domain continual benchmarks from previous works) where the aim is to perform well across many domains but the relevant data for each may exist more prominently in certain time periods of Common Crawl.

---

> ### Author Response · Authors · 2024-12-02
>
> Hi Reviewer YDQj! As the discussion period is coming towards an end, we wanted to gently reach out to see if you had a chance to consider our reply. We'd very much appreciate knowing if it helped address your initial concerns, and if so, whether you might be willing to reconsider your score. We'd also of course be happy to answer any additional questions you might have!

---

> > ### Comment · Reviewer_YDQj · 2024-12-03
> >
> > I thank authors' response on my comments.
> >
> > > TiC-Wiki dramatically increases the evaluation timespan of the original TemporalWiki from 4 months to 10 years. Also, as discussed in more detail in App. B.1, we used own parsing to handle format changes and maintain temporal consistency across Wikipedia/Wikidata dumps.
> >
> > The method you use is more like marginal improvements of existing methods.  rather than novel contribution.
> >
> > > TiC-CodeDocs is the first evaluation based on API documentation evolution, requiring version-aware parsers to process 32 major releases of NumPy and PyTorch while handling complex format changes.
> >
> > I recognize the novelty and contribution of the TiC-CodeDocs dataset.

---

> > > ### Author Response · Authors · 2024-12-04
> > >
> > > Thank you for considering our response and for the additional reply! Since your latest comment covered just the novelty of evaluations, we hope to take this as an indication that the other initial concerns you had have been addressed, though please let us know if otherwise!
> > >
> > > Now regarding evaluations, we first appreciate you recognizing the novelty of TiC-CodeDocs. As for TiC-Wiki, we do not see the main novelty lying in the evaluation construction, but rather its usage in the broader TiC-LM setup, which importantly differs from TemporalWiki (and other previous works) in terms of the generality of the training data (CC instead of perfectly aligned Wikipedia dumps) and timespan captured (11 years instead of 4 months). We believe both of these factors reflect important practical realities of LLM pre-training and in this context, TiC-Wiki led to unique insights such as the fact that peak performance on a certain month is often *years* after that month of TiC-CC is seen.

---

### Official Review · Reviewer_9XXG · 2024-11-01

**Soundness:** 3
**Presentation:** 2
**Contribution:** 2
**Rating:** 5
**Confidence:** 4

**Summary:**

This paper introduces a novel dataset for continual pretraining of large langauge models that significantly expands the scale of existing datasets in the field. The provided dataset may be valuable for future studies in the continual pretraining. However, the insights are limited. The experimental results are not well-explained.

**Strengths:**

1. **Impressive Dataset Size**: The proposed dataset significantly surpasses existing datasets in scale, providing a robust foundation for research.

2. **Rigorous Experimental Setting**: The experimental design is thorough and well-structured, ensuring reliable results.

3. **Comprehensive Review of Related Work**: The discussion of related work is thorough and effectively contextualizes the research within the field.

**Weaknesses:**

1. **Limited Insights**: The paper does not provide new insights into continual pretraining. The experimental results indicate that cyclic learning rate schedules, data replay, and regularization methods are insufficient for maintaining both strong in-domain performance and reduced forgetting. A clearer explanation of why each method fails under different conditions would enhance the understanding. The experimental section reads more like a report than an analysis, which limits its usefulness for future researchers.

2. **Unconvincing Experiments**: The performance of EWC is heavily influenced by the weight of the regularization term, creating a trade-off between plasticity and stability. There is a lack of detailed discussion regarding the hyperparameter analysis of EWC. While Appendix C mentions that “λ = 10^7 performed best when tuning between 10^1 and 10^9,” the varying performance of EWC across different evaluation datasets (as shown in Table 3) raises concerns. It’s possible for EWC to perform well on one dataset while underperforming on another. This issue extends beyond EWC to other methods as well, suggesting a sensitivity to hyperparameters that warrants further exploration.

3. **Poor Presentation**: The overall presentation, particularly in the experimental section, is lacking. Important analyses are often relegated to figure captions, such as those for Figure 4 and Table 2, which detracts from the clarity of the findings.

**Questions:**

Please refer to limitation

---

> ### Author Response · Authors · 2024-11-24
> **Response to Reviewer 9XXG [1/2]**
>
> Thank you for your time and thoughtful feedback! We hope to address your concerns and questions below.
>
> **Limited Insights.** Generally, we see the unique scale and scope of the TiC-LM as one of its primary contributions. To our knowledge, it is the first CL benchmark that captures two key aspects of LLM pre-training : (1) the time-evolving training data is generic web-data rather than single domains such as Wikipedia/Tweets/News; (2) the evaluations also span a variety of domains, reflecting the goal of pre-training to be useful in many possible contexts. As such, we believe that even takeaways about methods which re-affirm existing works remain non-trivial. At the same time, we’d also like to highlight some takeaways that uniquely show up in our setting:
>
> * *Optimization methods:*
>    * We found it necessary to make the maxLR in each continual round at least 30x smaller than for the initial pre-training (see the expanded App. C). This is in contrast to recipes from Ibrahim et al.; 2024 and Gupta et al., 2023, who re-warm to up to a similar maxLR or Parmar et al., 2024 who start from the minLR of the pre-trained model and cool down further. We believe this is because their setups involve only 2 or 3 (roughly) equally sized training rounds and distribution shifts across data quality / language rather than many (likely more gradual) temporal shifts.
>    * Since AR schedules roughly decay maxLR at a 1/t rate across rounds, they naturally result in a similar (though slightly better) performance profile to using a constant maxLR = minLR since our setup has >100 rounds.
>
> * *Replay methods:* While other works have experimented with replay and found it effective for avoiding forgetting, our results additionally shed light on important challenges that arise when continually pre-training over long timeframes:
>
>     * *How should replay be implemented when there are many timesteps (>100) to balance?* We find that $\alpha_t = 1/2$ offer much better plasticity than $\alpha_t = 1/t$, in contrast to TiC-CLIP which finds that different replay strategies perform similarly but only involves an order of magnitude fewer training rounds. Even for $\alpha_t = 1/2$, we find that while it can offer good balance between Backward and ID for a few years, it can result in increasingly reduced plasticity in the longer timescales captured by TiC-CC.
>     * *How does replay affect different domains when training on generically web-data?* In contrast to prior works that train and evaluate on single domains (e.g., Jang et al; 2021), here we find that different domains change at different rates and that certain time periods of Common Crawl can be crucial for specific evaluations (e.g TiC-StackE-Math and TiC-NumPy v.s. TiC-Stackoverflow and TiC-Pytorch).
>
> * *Regularization methods:* previous works that trained larger models often did not test LwF/EWC (perhaps due to the implementation challenges that we also faced, e.g. combining EWC with FSDP). In contrast to the more negative results from Garg et al.;, 2023 and Jin et al;, 2022, we found that EWC can be useful in our setup, resulting in the best performance on TiC-Wiki.
> * *A practical continual learning strategy:* since the submission we have also added an additional analysis (see Sec 6.3) of the practical utility of our current methods. We find that when scaling up Replay ($\alpha_t=1/2$) + AR for 2x more training tokens, it can become competitive to re-training Oracle methods every two years despite still using 62% less total compute.
>
> We have modified our manuscript (Sec 6.2, Sec 6.3, and App. C) to either add or make these points more explicit.

---

> ### Author Response · Authors · 2024-11-24
> **Response to Reviewer 9XXG [2/2]**
>
> **Unconvincing Experiments / Hyperparameter Sensitivity.** Since the submission, we have expanded our discussion about hyperparameters in App. C. While we shared details about our hyperparameter search procedure (which we wanted to make realistic by using only the first 10 months of TiC-CC), we agree that showing some results for the different configurations tested would be helpful. Given the compute costs of each continual run, we unfortunately were not able to complete full runs for all the methods/configurations explored during tuning. That being said, we do now include several results on the tuning months as well as a comparison to full results for Cyclic Cosine. For EWC specifically, we copy the table below, where we observe  $\lambda$ needs to be set quite high to have any noticeable effect but also enters a range where you trade-off backwards and ID performance. Our choice of $\lambda = 10^7$ strictly outperformed other choices except $10^6$ which had slightly better ID performance but considerably worse backward transfer. Going forwards, we are happy to further expand our study of hyperparameters, such as by training more tuning configurations to completion!
>
> | $\lambda_{\text{EWC}}$ || TiC-CC (Tuning Months) ||
> |:--------:|:------------------:|:----:|:---------:|
> |          | Backward | ID | Forward |
> | 10⁰ | 0.0025 | 0.0050 | 0.0394 |
> | 10¹ | 0.0025 | 0.0050 | 0.0394 |
> | 10⁴ | 0.0025 | 0.0050 | 0.0394 |
> | 10⁵ | 0.0025 | 0.0049 | 0.0394 |
> | 10⁶ | 0.0021 | **0.0047** | 0.0391 |
> | 10⁷ | **0.0013** | 0.0050 | **0.0389** |
> | 10⁸ | 0.0107 | 0.0178 | 0.0462 |
> | 10⁹ | 0.0286 | 0.0400 | 0.0586 |
>
>
>
> **Poor Presentation.** Thank you for pointing this out. In our revised manuscript, we have modified the Experiments section to make the main text less reliant on the captions and better highlight the key takeaways in relation to existing works. We also welcome any additional suggestions you might have!
>
> **References**
>
> [1] Adam Ibrahim, Benjamin Thérien, Kshitij Gupta, Mats L. Richter, Quentin Gregory Anthony, Eugene
> Belilovsky, Timothée Lesort, and Irina Rish. *Simple and scalable strategies to continually pre-train
> large language models.* TMLR 2024.
>
> [2] Kshitij Gupta, Benjamin Thérien, Adam Ibrahim, Mats Leon Richter, Quentin Gregory Anthony,
> Eugene Belilovsky, Irina Rish, and Timothée Lesort. *Continual pre-training of large language
> models: How to re-warm your model?* In Workshop on Efficient Systems for Foundation Models @
> ICML2023.
>
> [3] Jupinder Parmar, Sanjev Satheesh, Mostofa Patwary, Mohammad Shoeybi, and Bryan Catanzaro.
> *Reuse, don’t retrain: A recipe for continued pretraining of language models.* arXiv preprint
> arXiv:2407.07263, 2024
>
> [4] Joel Jang, Seonghyeon Ye, Changho Lee, Sohee Yang, Joongbo Shin, Janghoon Han, Gyeonghun Kim,
> and Minjoon Seo. *Temporalwiki: A lifelong benchmark for training and evaluating ever-evolving
> language models.* EMNLP 2022.
>
> [5] Saurabh Garg, Mehrdad Farajtabar, Hadi Pouransari, Raviteja Vemulapalli, Sachin Mehta, Oncel
> Tuzel, Vaishaal Shankar, and Fartash Faghri. *Tic-clip: Continual training of clip models.* In
> The Twelfth International Conference on Learning Representations (ICLR), 2024.
>
> [6] Xisen Jin, Dejiao Zhang, Henghui Zhu, Wei Xiao, Shang-Wen Li, Xiaokai Wei, Andrew O. Arnold,
> and Xiang Ren. *Lifelong pretraining: Continually adapting language models to emerging corpora.* NACCL 2022.

---

> > ### Comment · Reviewer_9XXG · 2024-11-26
> >
> > Thank you for your response. However, I remain concerned about the technical contributions of this paper beyond the dataset resource it provides. The experimental section reads more like a descriptive experimental report. For instance, neither Table 2 nor Table 3 demonstrates that any particular strategy consistently outperforms others across different settings. Moreover, setting $\alpha_t=1/2$ seems trivial. This hyperparameter is likely influenced by both the ratio of data from new tasks versus old tasks and the degree of overlap between new and old tasks. Consequently, I find it difficult to identify a practical strategy for continual pretraining on other datasets.
> >
> > Additionally, I believe it is necessary to highlight the modifications in the manuscript using a different color to improve clarity. Based on the reasons stated above, I will maintain my current score.

---

> ### Author Response · Authors · 2024-11-28
>
> Thank you for considering our response. We appreciate the chance for continued discussion and hope to address the points from your latest reply below:
>
> > The experimental section reads more like a descriptive experimental report...Additionally, I believe it is necessary to highlight the modifications in the manuscript using a different color to improve clarity.
>
> During the rebuttal period and after further integrating your feedback, we've re-formatted Sections 6.1 and 6.2 to better highlight the novel takeaways and added more discussion about their relationships to existing work. In the latest upload, we have also marked the new/changed text in blue, (apologies if it was confusing earlier). We hope these changes help address your concerns about presentation and of course would welcome any additional suggestions!
>
> > Neither Table 2 nor Table 3 demonstrates that any particular strategy consistently outperforms others across different settings.
>
> We see the different trade-offs between methods as an important result reflecting the inherent and *realistic* challenges of trying to continually pre-train LLMs from general web-data while also aiming to achieve strong performance across many domains. Because different domains evolve at different rates (and may be over/under-represented in different months of CC), the natural trade-offs between forgetting/plasticity that different methods (e.g., replay) induce will lead to corresponding trade-offs across evaluations. We believe due to its novel scope, **our benchmark is the first to capture and highlight these realistic challenges (e.g., compared to older efforts that trained and evaluated on single domains.)** Furthermore, we find that despite these difficulties, we are still able to identify a continual method that, while not the best individually on every evaluation, demonstrates significant practical utility overall. As mentioned in our first response, our new results in Sec 6.3 show that  Replay ($\alpha_t = 1/2$) + AR results in a competitive alternative to periodically re-training Oracle models while being 2.6x more compute efficient.
>
> > Moreover, setting $\alpha_t = 1/2$ seems trivial. This hyperparameter is likely influenced by both the ratio of data from new tasks versus old tasks and the degree of overlap between new and old tasks. Consequently, I find it difficult to identify a practical strategy for continual pre-training on other datasets.
>
> In our study, we chose to try $\alpha_t = 1/2$ and $\alpha_t = 1/t$ exactly because they were (a) simple/intuitive strategies that did not require careful tuning (which indeed might risk overfitting to our setup); (b) found to be effective choices in TiC-CLIP (there named Cumulative-Equal and Cumulative-All). As mentioned, our new results in Sec 6.3 show that despite its simplicity, $\alpha_t = 1/2$ can already result in 2.6x practical efficiency gains. Further, even within the 11 years covered by our dataset, it is quite likely that the data changes more/less quickly in some time periods. This suggests that $\alpha_t = 1/2$'s effectiveness may already show some robustness to natural variations in how fast data evolves.
>
> At the same time, we do agree that varying $\alpha_t$ based upon your data is something that could yield even better results. In the remainder of the discussion period, we are happy to try running with other fixed settings of $\alpha_t$, while more complex/adaptive strategies for setting $\alpha_t$ would certainly be interesting directions for future work!

---

> > ### Comment · Reviewer_9XXG · 2024-11-29
> >
> > Thank you for your response! I believe the presentation has improved significantly after the rebuttal.
> >
> > I have one additional question: In the captions of Table 2 and Table 3, it states, "Bold values are within one standard deviation." Could you clarify why this is the case? If the intention is to highlight statistically significant improvements, the bold values should ideally represent the best results that outperform the second-best results by at least three standard deviations, rather than just one.

---

> > > ### Author Response · Authors · 2024-12-01
> > >
> > > Thank you very much for acknowledging the improved presentation, as well as for your original suggestions regarding it!
> > >
> > > Re: bolding, we chose to highlight every method that achieved within 1 std (68% CI assuming normality) of the best result instead of only the best result (which is the more common practice in papers) precisely to try to avoid over-claiming about specific methods. Using 3 stds (99.7% CI) would be (quite a bit) stricter but we note that while a few additional highlights would be added to some evals, the key takeaways would not change. In future revisions, we'd be happy to consider additionally formatting (e.g., underline, color) results within 2 or 3 stds to paint an even more detailed picture of the gaps between methods.

---

> > > > ### Comment · Reviewer_9XXG · 2024-12-01
> > > >
> > > > Thanks for the clarification. While I think the new insights are limited, I think this is probably acceptable for a paper submitted to the "Datasets and Benchmarks" track. Given the improvements in presentation and the additional results, I would like to increase the score to 5.

---

> > > > > ### Author Response · Authors · 2024-12-02
> > > > >
> > > > > Thank you for engaging in discussions with us and for raising your score! For completeness, as mentioned in a previous response, we still plan to share some additional results regarding alternative choices for $\alpha_t$ (that are currently running but should finish before the end of the discussion period). In general, if there are any other ways to further resolve your remaining concerns, please let us know!

---

> ### Author Response · Authors · 2024-12-04
>
> To follow up on our promise to try different $\alpha_t$ values, we have now completed a set of runs for $\alpha_t \in \\{0.1, 0.3, 0.5, 0.7, 0.9\\}$. By doing a controlled sweep over the amount of replay used, we find that:
>
> * Different evaluations benefit from more/less replay (reinforcing some of our earlier takeaways about Replay)
>    * On TiC-CC subsets, we see trade-offs between Backward and ID. More replay helps on Backward but does worse on ID.
>    * On downstream evals, the optimal $\alpha_t$ varies. Lower $\alpha_t$ (more replay) helps more on the slower-changing domains (e.g., TiC-CodeDocs-NumPy) and higher $\alpha_t$ on faster moving ones (e.g., TiC-CodeDocs-PyTorch)
> * Using $\alpha_t = 1/t$ is generally sub-optimal, as we now see it is typically dominated by $\alpha_t = 0.1$ on all three metrics. As mentioned earlier, this is a notable difference from the findings in TiC-CLIP, which pushes forward the $\alpha_t = 1/t$ variant of replay, which we believe **scales more poorly here given the >10x larger amount of timesteps captured by TiC-LM.**
> * Given the trade-off across evals, amongst fixed choices of $\alpha_t$, **using a middling value like $\alpha_t = 0.5$ (or slightly higher) seems to be a reasonable practical recommendation** to avoid performing poorly on any one evaluation/domain. Of course, there's certainly room for future work to develop more sophisticated replay methods that are adaptive (e.g., determining $\alpha_t$. based upon monthly loss histories) and/or data-dependent (e.g., replaying only certain subsets of older data)!
>
>
>
> |   | TiC-CC   |  |    &#124;      |   TiC-Wiki-Diff |  |        &#124;        | TiC-CD-Numpy |  |    &#124;   |   TiC-CD-PyTorch   |   |
> |---|:--------:|:------:|:-------:|:---------------:|:-------------:|:-------------:|:------------:|:------------:|:----:|:------------------:|:-------------:|
> |   | Backward |   ID   | &#124; | Backward |      ID       | &#124; | Backward |         ID         | &#124; | Backward |         ID           |
> | $\alpha_t=1/t$ | 0.023 | 0.074 | &#124; | 0.038 | 0.063 | &#124; | 0.054 | 0.046 | &#124; | 0.175 | 0.138 |
> | $\alpha_t=0\.1$ | **0.022** | 0.066 | &#124; | 0.036 | 0.061 | &#124; | **0.042** | **0.045** | &#124; | 0.165 | 0.146 |
> | $\alpha_t=0\.3$  | **0.022** | 0.052 | &#124; |0.034 | 0.058 | &#124; | 0.056 | 0.059 | &#124; | 0.132 | 0.111 |
> | $\alpha_t=0\.5$  | 0.024 | 0.042 | &#124; |**0.032** | 0.055 | &#124; |0.058 | 0.066 | &#124; | 0.099 | 0.069 |
> | $\alpha_t=0\.7$ | 0.028 | 0.035 | &#124; |**0.032** | 0.053 | &#124; | 0.066 | 0.083 | &#124; | 0.090 | 0.055 |
> | $\alpha_t=0\.9$  | 0.039 | 0.029 | &#124; | 0.033 | **0.052** | &#124; | 0.069 | 0.078 | &#124; | 0.070 | 0.035 |
> | $\alpha_t=1\.0$  | 0.072 | **0.027**  | &#124; | 0.033 | **0.052** | &#124; | 0.073 | 0.096 | &#124; | **0.057** | **0.025** |
>
>
> Here, we share a portion of the new results and in future revisions, while we plan to include the complete set into our manuscript in future revisions!

---

### Official Review · Reviewer_jWj5 · 2024-11-02

**Soundness:** 4
**Presentation:** 3
**Contribution:** 4
**Rating:** 8
**Confidence:** 4

**Summary:**

The paper presents a novel benchmark, TiC-LM, for evaluating continual pretraining of large language models (LLMs). The benchmark is structured around a dataset called TIC-CommonCrawl (TIC-CC), which aggregates 114 monthly snapshots from Common Crawl from 2013 to 2024. The authors simulate a continual training environment by sequentially feeding data from each monthly snapshot to an LLM and exploring various continual learning methods. The key contributions include evaluating the temporal relevance of language models over time, assessing forgetting and forward transfer across domains, and determining effective methods for balancing the retention of old knowledge with learning new information. This paper is a huge engineering project that provides a benchmark that thorougly evaluates existing methods for continual LLM pretraining and can allow the field to quantify progress in developing better methods.

**Strengths:**

Originality: The work introduces a unique benchmark for continual language model pretraining.
Quality: The authors have employed a thorough and robust experimental methodology, including multiple baselines and diverse continual learning techniques, demonstrating the rigor of their approach.
Clarity: The problem is well-motivated, with clear research questions and systematically presented results.
Significance: The findings provide insights into temporal dynamics in LLMs, especially the trade-offs between data replay and learning rate adjustments for preserving knowledge across time. This work has strong potential to serve as a reference benchmark for the community.

**Weaknesses:**

Lack of novel findings: The paper lacks novel findings not explored in previous works. However, it does improve upon previous benchmarks and can serve as a better indicator for quantifying progress towards developing better methods for this problem.
Complexity of Setup: The continual pretraining process as described may be challenging for broad adoption without substantial compute resources, making it difficult for smaller research groups to replicate the study.
Hyperparameter Sensitivity: The paper does not fully address how sensitive different strategies (like cyclic cosine schedules) are to hyperparameters, which could impact reproducibility.

**Questions:**

How sensitive are the results to hyperparameter choices, especially for cyclic learning rate schedules and replay ratios? Additional insights here would be helpful for reproducibility.
Do the authors plan to release the TIC-LM dataset, and if so, would it include any pre-trained checkpoints for baselines to facilitate easier reproduction of results?

---

> ### Author Response · Authors · 2024-11-24
> **Response to Reviewer jWj5 [1/2]**
>
> Thank you for your time and thoughtful feedback! We hope to address your concerns below.
>
> **Novelty of findings.** We appreciate you recognizing the uniqueness and potential of our benchmark! Regarding our findings, we believe that given the novel scale/scope of our setting (as you pointed out) even takeaways about methods which re-affirm existing works are non-trivial. At the same time, we’d also like to highlight some takeaways that uniquely show up in our setup:
>
> * *Optimization methods:*
>    * We found it necessary to make the maxLR in each continual round at least 30x smaller than for the initial pre-training (see expanded App. C). This is in contrast to recipes from Ibrahim et al.; 2024 and Gupta et al., 2023, who re-warm to up to a similar maxLR or Paramar et al., 2024 who start from the minLR of the pre-trained model and cool down further. We believe this is because their setups involve only 2 or 3 (roughly) equally sized training rounds and distribution shifts across data quality / language rather than many (likely more gradual) temporal shifts.
>    * Since AR schedules roughly decay maxLR at a 1/t rate across rounds, they naturally result in a similar (though slightly better) performance profile to using a constant maxLR = minLR since our setup has >100 rounds.
>
> * *Replay methods:* While other works have experimented with replay and found it effective for avoiding forgetting, our results additionally shed light on important challenges that arise when continually pre-training over long timeframes:
>
>     * *How should replay be implemented when there are many timesteps (>100) to balance?* We find that $\alpha_t = 1/2$ offer much better plasticity than $\alpha_t = 1/t$, in contrast to TiC-CLIP which finds that different replay strategies perform similarly but only involves an order of magnitude fewer training rounds. Even for $\alpha_t = 1/2$, we find that while it can offer good balance between Backward and ID for a few years, it can result in increasingly reduced plasticity in the longer timescales captured by TiC-CC.
>     * *How does replay affect different domains when training on generically web-data?* In contrast to prior works that train and evaluate on single domains (e.g., Jang et al; 2021), here we find that different domains change at different rates and that certain time periods of Common Crawl can be crucial for specific evaluations (e.g TiC-StackE-Math and TiC-NumPy v.s. TiC-Stackoverflow and TiC-Pytorch).
>
> * *Regularization methods:* previous works that trained larger models often did not test LwF/EWC (perhaps due to the implementation challenges that we also faced, e.g. combining EWC with FSDP). In contrast to the more negative results from Garg et al.;, 2023 and Jin et al;, 2022, we found that EWC can be useful in our setup, resulting in the best performance on TiC-Wiki.
> * *A practical continual learning strategy:* since the submission we have also added an additional analysis (see Sec 6.3) of the practical utility of our current methods. We find that scaling up Replay ($\alpha_t=1/2$) + AR for 2x more training tokens, it can become competitive to re-training Oracle methods every two years despite still using 62% less total compute.
>
> We have updated our manuscript (Sec 6.1, 6.2, Sec 6.3, and App. C) to either add or discuss these points more explicitly!
>
> **Complexity of setup.** We plan to share all code for reproducing TiC-CC from Common Crawl as well as the training code for all methods. We will also provide model checkpoints for the initialization, oracle, and various continual runs. For lower-resource research groups, our scripts can be used to generate smaller subsets for continual pre-training with fewer tokens/months/model parameters while our pre-trained checkpoints can also allow for evaluating methods when using only a few months of additional training. We’re happy to provide some additional experiments in these regimes in future revisions.

---

> ### Author Response · Authors · 2024-11-24
> **Response to Reviewer jWj5 [2/2]**
>
> **Sensitivity to hyperparameters.** Since the submission, we have expanded our discussion about hyperparameters in App. C. While we previously shared higher-level details about our search procedure (which tunes on the first 10 continual months), we agree that showing results for different configurations would be helpful. Given the compute costs of each continual run, we were not able to complete full runs for all the methods/configurations we previously only ran for the tuning months. However, we were able to do so for the example you asked about re: LRs for Cyclic Cosine (copied below) and also included results on the tuning months for some other methods, observing for instance:
>
> * Max LR in each cyclic schedule needs to be low enough but once it reaches that point, lowering it further can trade-off backwards and ID performance. However, even a very small or AR-decayed max LR does not offer better backwards transfer compared to using replay.
> * EWC’s regularization weight needs to be set quite high but also enters a regime where you start to trade-off backwards and ID performance. Our choice of $\lambda = 10^7$ strictly outperformed most other choices except $10^6$ which had slightly better ID performance but considerably worse backward transfer.
> * For replay, we have not yet tried anything beyond the two configurations we mentioned ($\alpha_t = 1/t$ and $\alpha_t = 1/2$), with the takeaways already described in the above respone.
>
> Going forwards, we are happy to further expand our study of hyperparameters (e.g., training more tuning configurations to completion)!
>
> | Max LR || TiC-CC (Tuning Months) ||| TiC-CC (All Months) ||
> |-----------|:------------------:|:----:|:---------:|:---------------:|:----:|:---------|
> |        | Backward | ID | Forward | Backward | ID | Forward |
> | 1e-3 | 0.103 | 0.086 | 0.118 | 0.197 | 0.083 | 0.209 |
> | 3e-4 | 0.019 | 0.016 | 0.051 | 0.125 | 0.041 | 0.178 |
> | 1e-4 | **0.002** | **0.005** | **0.039** | 0.072 | **0.027** | **0.161** |
> | 5e-5 | **0.002** | 0.006 | **0.039** | 0.062 | 0.034 | 0.163 |
> | 3e-5 | 0.004 | 0.009 | 0.040 | 0.060 | 0.042 | 0.165 |
> | AR Schedule | **0.002** | 0.008 | 0.043 | **0.058** | 0.040 | 0.166 |
>
> **References**
>
> [1] Adam Ibrahim, Benjamin Thérien, Kshitij Gupta, Mats L. Richter, Quentin Gregory Anthony, Eugene
> Belilovsky, Timothée Lesort, and Irina Rish. *Simple and scalable strategies to continually pre-train
> large language models.* TMLR 2024.
>
> [2] Kshitij Gupta, Benjamin Thérien, Adam Ibrahim, Mats Leon Richter, Quentin Gregory Anthony,
> Eugene Belilovsky, Irina Rish, and Timothée Lesort. *Continual pre-training of large language
> models: How to re-warm your model?* In Workshop on Efficient Systems for Foundation Models @
> ICML2023.
>
> [3] Jupinder Parmar, Sanjev Satheesh, Mostofa Patwary, Mohammad Shoeybi, and Bryan Catanzaro.
> *Reuse, don’t retrain: A recipe for continued pretraining of language models.* arXiv preprint
> arXiv:2407.07263, 2024
>
> [4] Joel Jang, Seonghyeon Ye, Changho Lee, Sohee Yang, Joongbo Shin, Janghoon Han, Gyeonghun Kim,
> and Minjoon Seo. *Temporalwiki: A lifelong benchmark for training and evaluating ever-evolving
> language models.* EMNLP 2022.
>
> [5] Saurabh Garg, Mehrdad Farajtabar, Hadi Pouransari, Raviteja Vemulapalli, Sachin Mehta, Oncel
> Tuzel, Vaishaal Shankar, and Fartash Faghri. *Tic-clip: Continual training of clip models.* In
> The Twelfth International Conference on Learning Representations (ICLR), 2024.
>
> [6] Xisen Jin, Dejiao Zhang, Henghui Zhu, Wei Xiao, Shang-Wen Li, Xiaokai Wei, Andrew O. Arnold,
> and Xiang Ren. *Lifelong pretraining: Continually adapting language models to emerging corpora.* NACCL 2022.

---

> > ### Author Response · Authors · 2024-12-02
> >
> > Hi Reviewer jWj5! As the discussion period is coming towards an end, we wanted to gently reach out to see if you had a chance to consider our reply. We'd very much appreciate knowing if it helped address your initial concerns, and if so, whether you might be willing to reconsider your score. We'd also of course be happy to answer any additional questions you might have!

---

### Official Review · Reviewer_x8ae · 2024-11-03

**Soundness:** 3
**Presentation:** 3
**Contribution:** 2
**Rating:** 6
**Confidence:** 4

**Summary:**

This paper studies the continual training of LLMs. In general LLMs are pretrained on a time snapshot of the data and do not get retrained very often. As a result, as new data become available, the knowledge of model become outdated and does not cover new information. This paper offers benchmarks to evaluate this effect and also offer some solutions on how to update the model with the new data while avoiding forgetting the previous data.

**Strengths:**

- the paper is well written and the datasets, steps, and evaluations are clearly explained
- the paper tackles an important problem: how to continuously update LLMs without needed to fully retrain

**Weaknesses:**

My main concern is the contribution of the paper. Although the authors have tried to extensively quantify the effect of model not being trained on the new data, it is a known fact that training in distribution is always better. So if we evaluate the model on the same data from future, there would be degradation. The more interesting question is the remediation recipe for this problem and more importantly how the model can be adopted to new domains. Authors study few methods of updating the model and show some work better than the other however, all lacking behind the oracle and they leave it to future work. I believe emphasizing on the benchmark by itself does not bring enough novelty to warrant acceptance.

**Questions:**

- Authors mention that they do only fuzzy deduplication within each shard. How do they guarantee that there is no leakage between the shards especially for the evaluation sets?

---

> ### Author Response · Authors · 2024-11-24
> **Response to Reviewer x8ae**
>
> Thank you for your time and thoughtful feedback! We hope to address your two main concerns below.
>
> **Novelty + Closing gap to Oracle.** We agree the ultimate goal is to find practical continual learning methods. To that end, we see our more realistic benchmark, which is the first of appropriate scale/scope for LLM *pre-training* (i.e., using generic web-crawled training data and evaluations from multiple domains), as an important pre-requesite that is missing from the current literature.
>
> Regarding our methods not matching the Oracle, we note that this Oracle is quite strong for two reasons: (1) while it is compute matched with full continual runs, it has seen considerably more tokens compared to most continual checkpoints in our eval matrices (up to 2x for earliest months); (2) it would not actually be obtainable until the very last month; if one wanted to consistently have up-to-date models re-trained like the Oracle, the costs would increase linearly with the number of desired checkpoints. Since the submission, we have run additional experiments to better contextualize the cost effectiveness of our current continual strategies relative to re-training Oracles:
>
> * *Token-matched oracles:* We consider a baseline which is a _series_ of 7 oracles based on different cutoff dates. Each is token matched with the existing 220B continual runs based upon cutoff date (e.g., a 2019-01 oracle is trained on all data up to 2019-01 for the same number of total tokens seen by a continual run’s 2019-01 checkpoint). We use the cutoffs {2013-05, 2015-01, 2017-01, 2019-01, 2021-01, 2023-01, 2024-07} which combined see 1.16T tokens. When measuring sub-optimality for continual checkpoints against this series, we now subtract the performance of the most recent Oracle (instead of always 2024-07).
> * *Scaled-up continual runs:* We try scaling up continual runs by increasing the budget allocated to each month in the continual phase by 2x and 3x (while keeping the same initialization trained on 110B tokens). These runs see 330B and 440B total tokens.
>
> We add these new results in Sec 6.3 of our updated manuscript (and copied below), where we report the average of the Backwards and ID elements of the matrices for different evaluations. Overall, we observe that despite requiring about 62% less compute, the longest 440B continual runs are fairly competitive with the series of the Oracles, surpassing it on all TiC-CC and TiC-Wiki evaluations and closing the gap on most others. This demonstrates that our continual methods already showcase practical relevance, though we hope further development using our benchmark can yield even larger efficiency gains! In future revisions, we're happy to expand this analysis (e.g., obtain results for scaling up other continual methods).
>
> | Method | Tokens | TIC-CC | TIC-CC-Wiki | TIC-CC-News | TIC-wiki-Diff | TIC-Wiki-Unchanged | TIC-Stackoverflow | TiC-StackE-Cat7 | TiC-CodeDocs-PyTorch | TiC-CodeDocs-NumPy |
> |--------|---------|--------|------------|------------|--------------|-------------------|------------------|---------------|-------------|------------|
> | Replay | 220B | 0.024 | 0.019 | 0.017 | 0.013 | 0.017 | 0.034 | 0.028 | 0.052 | 0.047 |
> | Replay + AR | 220B | 0.027 | 0.017 | 0.014 | 0.014 | 0.016 | 0.027 | 0.022 | 0.082 | 0.023 |
> | Replay | 330B | 0.011 | 0.009 | 0.010 | 0.001 | 0.006 | 0.014 | 0.020 | 0.003 | 0.035 |
> | Replay + AR | 330B | 0.008 | 0.003 | 0.002 | -0.001 | -0.001 | 0.017 | 0.014 | 0.040 | 0.024 |
> | Replay | 440B | 0.004 | 0.004 | 0.007 | -0.004 | 0.000 | **0.007** | 0.150 | **-0.025** | **0.018** |
> | Replay + AR | 440B | **-0.002** | **-0.006** | **-0.005** | **-0.009** | **-0.010** | 0.015 | **0.008** | 0.013 | 0.025 |
>
> **Leakage between training and evaluation sets in TiC-CC.** We agree that this is a potential concern, which we also noted previously in App. A. To clarify, we held out test examples at a document-level, but certainly the contents of particular pages may overlap. To test whether potential leakages affect our results, we have now also run an additional set of TiC-CC evaluations on decontaminated versions of the evaluation sets, where we used the same Bloom Filter approach to deduplicate the evaluation sets (except we now pre-populate the Bloom Filter with the corresponding training set). We then plot the two loss values (relative to the Oracle) given by the original and decontaminated evaluation sets for all (method, checkpoint, evaluation month) triplets in Figure 6 in Appendix A . Overall, we observe that the results are highly correlated, suggesting that data leakages do not affect our conclusions.

---

### Author Response · Authors · 2024-11-26

We thank all the reviewers for their time and effort during the review process! In general, we really appreciate the positive feedback on the practical relevance and potential of our benchmark (from all reviewers) as well as the constructive feedback regarding how our submission can been improved.

We have responded to the concerns from each review individually in their respective comments sections. We have also updated our manuscript as a result of this feedback, with new changes in blue. Here, we list the main changes:
* We have re-written Sec 6.1 and Sec 6.2 to improve the presentation in the main text (9XXG), better clarify the novel takeaways in relation to existing work (jWj5, 9XXG), and to elaborate on the different observations between evaluations such as between TiC-CC and TiC-Wiki (9XXG, YDQj).
* In Sec 6.3, we have added results that aim to better contextualize the utility/efficiency of current methods in relation to a baseline that acknowledges the full costs of maintaining a *series* of Oracle models (re-trained every two years). We find that scaling up our Replay ($\alpha_t = 1/2$) methods to 440B total tokens starts to become competitive with re-training Oracles despite using 62\% less overall compute. We hope this helps address concerns about the performance of existing methods (x8ae, 9XXG) and provides another novel insight that arises from our benchmark (x8ae, jWj5, 9XXG).
* In App. A, we have added a new plot (Figure 6) to address the concerns about potential data leakage between monthly TiC-CC training and evaluation sets  (x8ae). This shows that the relative losses between continual checkpoints and the Oracle are extremely correlated when we measure them on the current evaluation sets v.s. further decontaminated versions of them
* In App. C, we have expanded our discussion of hyperparameter tuning, sharing additional results on the 10-month tuning sets for various methods and providing additional analysis for sensitivity to specific choices like LR for Cyclic Cosine (jWj5) and regularization weight for EWC (9XXG).

---

### Meta-Review · Area_Chair_NXQ8 · 2024-12-22

**Metareview:**

### Claims and Findings:
 - This paper introduces a large-scale benchmark for continually training language models and evaluates continual learning strategies such as rate schedules, data replay, and regularization methods using this benchmark.
### Strengths:
   - The proposed benchmark significantly increases the size of current datasets.
   - The experimental design is well-structured and fairly comprehensive. Some empirical observations are useful, though may not be surprising.
### Weaknesses:
   - The methods for dataset curation and downstream evaluation are extension of the existent DataComp-LM approach and lack novelty.
   - The experiments on the benchmark yield limited new insights. Although there are some useful observations regarding training details, the benchmark does not provide inspiring or significant insights.
### Reasons for Decision:
   - Based on the identified weaknesses.

**Additional Comments On Reviewer Discussion:**

This paper has significantly benefited from the review process. The updated version shows considerable improvement in writing compared to the initial submission, prompting Reviewer 9XXG to raise their score from 3 to 5 after the rebuttal. However, the paper still falls short in novelty, in both the dataset construction method and the new insights gained from experimenting with such a large-scale dataset, which diminishes its overall contribution. This paper could benefit from another round of revision to address these issues.

---

### Decision · Program_Chairs · 2025-01-22

Reject